# Roadmap to the study of gene and protein phylogeny and evolution—A practical guide

**Florian Jacques** [1,2], **Paulina Bolivar**[1], **Kristian Pietras**[1], **Emma U. Hammarlund**[1,2]*

**1** Lund University Cancer Centre, Department of Laboratory Medicine, Lund University, Lund, Sweden,
**2** Lund Stem Cell Center, Department of Laboratory Medicine, Lund University, Lund, Sweden

* emma.hammarlund@med.lu.se

## Abstract

Developments in sequencing technologies and the sequencing of an ever-increasing number of genomes have revolutionised studies of biodiversity and organismal evolution. This accumulation of data has been paralleled by the creation of numerous public biological databases through which the scientific community can mine the sequences and annotations of genomes, transcriptomes, and proteomes of multiple species. However, to find the appropriate databases and bioinformatic tools for respective inquiries and aims can be challenging. Here, we present a compilation of DNA and protein databases, as well as bioinformatic tools for phylogenetic reconstruction and a wide range of studies on molecular evolution. We provide a protocol for information extraction from biological databases and simple phylogenetic reconstruction using probabilistic and distance methods, facilitating the study of biodiversity and evolution at the molecular level for the broad scientific community.

## Introduction

Living organisms are characterized by an astonishing array of phenotypes. This diversity is the result of billions of years of evolution, from the first primitive cells to modern cells and multicellular organisms, including bacteria, archaea, protists, plants, and animals. How organismal functions, adaptations, and diversifications are related can be studied through molecular evolution, a field that studies variation in the information content in the genetic material through time, namely the evolution of genes, proteins, or other markers such as ribosomal RNA, transposable elements, or other parts of the genome, that have common ancestry and are, therefore, homologous. Homology can result from speciation events, that create homologous genes in different species, or gene or genome duplication, that generate homologous genes in the same genome. Homologous genes in different species (*e.g.*, human and murine hemoglobin) are called orthologues, and homologous genes present in the same genome (*e.g.*, human hemoglobin and human myoglobin) are called paralogues. Gene homology resulting from horizontal transfer is called xenology. Phylogenetics and evolutionary biology study how homologues are related to each other to retrace their evolutionary history.

Evolutionary studies can be carried out in the context of comparative genomics between different species or, alternatively, in the context of population genetics, and compare molecular variation between the populations or individuals of a single species. In the first case, one considers long-term evolution and compares mutations (including substitutions) that have been fixed between different species. The second case considers short-term evolution and concerns

**Data Availability Statement:** Data are within the Supporting Information files.

**Funding:** EH received funding from the European Research Council (ERC) under the European Union's Horizon 2020 research and innovation

programme (grant agreement No 949538). The funder had and will not have a role in study design, data collection and analysis, decision to publish, or preparation of the manuscript.

**Competing interests:** The authors have declared that no competing interests exist.

the study of variation that is still segregating in a population. In both cases, the evolution of species is usually presented as a phylogenetic tree, a diagram displaying the evolutionary relationships between the sequences or taxa. The tools and methods for phylogenetic inference have become more complex over past four decades and their use can be challenging.

Molecular evolutionary studies aim at reconstructing the evolutionary histories and relationships of different taxa, genes or genomic components (e.g., transposable elements), as well as understanding the diverse mechanisms and factors underlying evolutionary change, such as mutation, selection, recombination, genetic drift, demographic processes, or biased gene conversion. For these purposes, the integration of novel genomic technologies with evolutionary studies are invaluable. For example, in systematics, the description of new species necessitates knowing how they are related to other species. In epidemiology, the emergence of new infectious diseases and antibiotic resistance requires studying genetic variation of infectious agents and identifying adaptive mutations leading to pathological conditions. A focus on the process of adaptation is also valuable in biological, agricultural and environmental sciences to, for example, protect endangered species or limit the spread of invasive species.

In recent years, technological advances in molecular biology, in particular the sequencing of DNA and RNA, has allowed for an exponential increase of available sequences of nucleotides and amino acids. In addition, these data are coupled with annotations regarding biological functions. The genomic, transcriptomic, and proteomic data is curated in specialized public databases, assets that are paralleled by development of new statistical methods and computational technology to study gene and protein functions and evolution. While generally accessible to biologists for studying molecular diversity and evolution, to sort and navigate through these resources can be challenging. Here, we outline a selection of molecular databases as well as bioinformatic tools and methods for retrieving sequences and reconstructing evolutionary history and processes. In doing so, we follow a recently published phylogenetic protocol [1]. We focus on databases that are maintained and popular. The aim is to provide a practical guide for beginners and more advanced explorers into protein and gene evolution. This is followed by a tutorial to the reconstruction of the evolution of two families of cell cycle-related proteins: P53 and cyclins/cyclin-dependent kinases (CDKs), over organismal history.

## Materials and methods

Data included in this study (sequences and accession numbers) are available in S1–S4 Files. There are no ethical or legal restrictions on sharing these data sets. The protocol described in this peer-reviewed article is published on protocols.io, https://protocols.io/view/road map-to-the-study-of-gene-and-protein-phylogeny-cknkuvcw and included as S5 File.

### Collecting genomic and proteomic information

Dozens of databases store sequences and other biological information about genes and proteins; for a complete list, see [2, 3]. These databases offer query tools to retrieve DNA or amino-acid sequences and other information such as gene architecture or protein structure. They also provide annotations with information about gene or protein properties such as function, polymorphism, activity and pathways, subcellular localization, and tissue expression (**Fig 1**).

### DNA databases

**GenBank** [4] and **Entrez** [5], both maintained at the National Centre for Biotechnology Information (**NCBI**) [6], store nucleotide sequences of all living organisms and, when applicable, their translation into protein, with biological annotation and supporting bibliography (**Table 1**). They include integrated search tools to retrieve sequences, structures, genetic

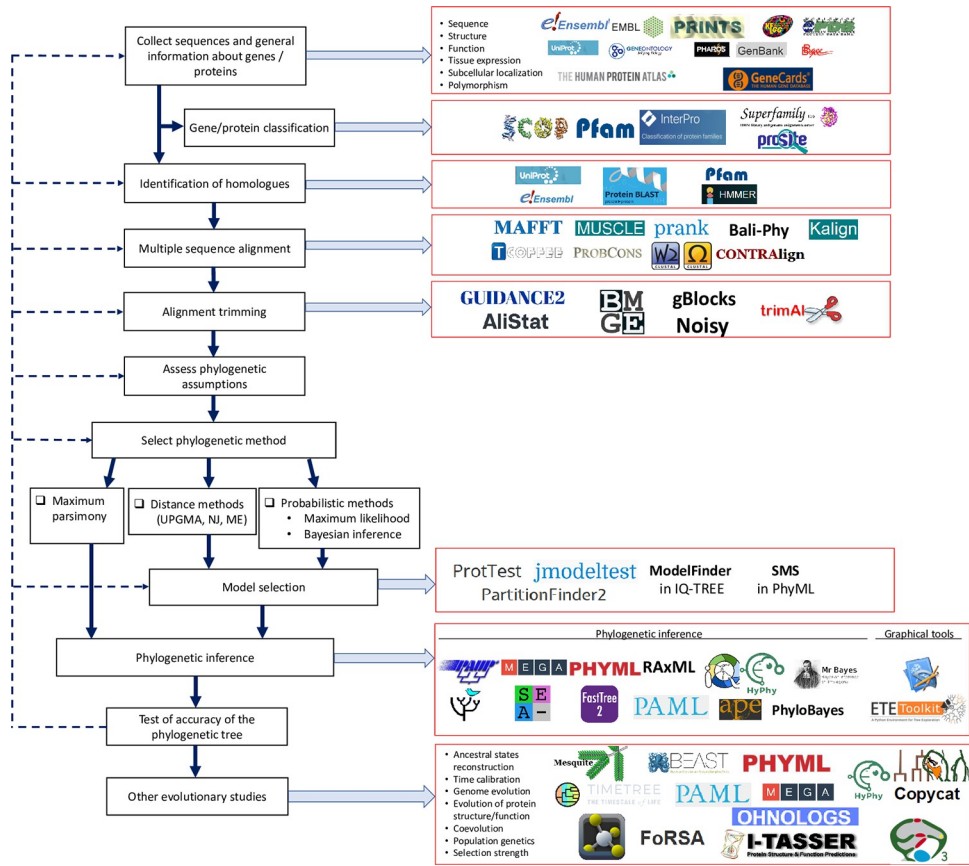

**Fig 1. Protocol for reconstructing the phylogeny and evolutionary history of genes and proteins using molecular databases and bioinformatic tools.** Solid arrows indicate the order of actions for the phylogenetic analysis and evolutionary studies. Dashed arrows indicate feedback loops that are needed during the process. A subset of available databases and bioinformatic programs are depicted in the figure. This roadmap is mostly based on a recently published phylogenetic protocol [1].

**Table 1. List of nucleic acid databases.**

| Database | Features | Link | References |
|---|---|---|---|
| BAR | Database of plant genes and proteins | http://bar.utoronto.ca/ | [15] |
| Bgee | Gene expression patterns | https://bgee.org/ | [8] |
| Ensembl | Genome browser of vertebrates, includes tools for identification of homology | https://www.ensembl.org/index.html | [7] |
| Entrez | Gene sequences and structures | https://www.ncbi.nlm.nih.gov/Web/Search/entrezfs.html | [5] |
| FlyBase | Genome and proteome of the model insect *D. melanogaster* | https://flybase.org/ | [11] |
| GeneCards | Human gene function, genomics, transcription factor binding sites and protein products | https://www.genecards.org/ | [9] |
| GenBank | Annotated DNA sequences | https://www.ncbi.nlm.nih.gov/genbank/ | [4] |
| NCBI | Collection of databases for molecular biology and medicine providing many bioinformatics tools and services | https://www.ncbi.nlm.nih.gov/ | [6] |
| PomBase | Genome and proteome of the model yeast *S. pombe* | https://www.pombase.org/ | [13] |
| TAIR | Genome and proteome of the model plant *A. thaliana* | https://www.arabidopsis.org/ | [14] |
| WormBase | Genome and proteome of the model nematode *C. elegans* | https://wormbase.org//#012-34-5 | [12] |
| Xenbase | Genome and proteome of the model amphibian *X. laevis* | http://www.xenbase.org/entry/ | [10] |

cartography and bibliography about genes [6]. **Ensembl** [7] is a genome browser that focuses on chordates and contains information about gene sequence and structure, expression, location on the chromosome, transcript variants, homologues, and gene ontologies. The browser is further expanded into specific databases for invertebrates, plants, fungi, protists, and bacteria in EnsemblMetazoa, EnsemblPlants, EnsembleFungi, EnsemblProtists and EnsemblBacteria. Ensembl is relevant for evolutionary analyses, comparative genomics, and population genetics studies. Data on gene expression patterns across animal species, including anatomical and embryonic information, is stored in the database **Bgee** [8]. **GeneCards** [9] stores information on human genes, including biological function, genomics, transcription factor binding sites and protein products, as well as assay products (e.g., siRNA, inhibitors or CRISPR products) and crosslinks to many other databases.

Particular model organisms that have provided extensive biological data are stored in organism-specific databases such as **XenBase** [10], **FlyBase** [11], and **WormBase** [12]. They include data concerning genomics, development, gene expression and variants of the amphibian *Xenopus laevis*, the fruit fly *Drosophila melanogaster*, and the nematode *Caenorhabditis elegans*, respectively. Data from the fission yeast *Schizosaccharomyces pombe* is stored in **PomBase** [13], which includes the complete genome, gene and protein sequences, and annotations. The Arabidopsis Information Resource (**TAIR**) [14] provides the complete genome sequence of the model plant *Arabidopsis thaliana* and information on gene sequence and structure, gene expression, protein sequence and literature. The Bio-Analytic Resource (**BAR**) [15] for plant biology also provides access to several plant-specific databases, including gene expression and protein tools such as the eFP Browser (https://bar.utoronto.ca/efp/cgi-bin/efpWeb.cgi) [16], that displays gene expression patterns in Arabidopsis, molecular markers and mapping and genomic tools.

Nucleotide sequences can be downloaded from these databases in the FASTA format (a text-based format for representing either nucleotide or amino-acid sequences, where nucleotides or amino acids are represented by single-letter codes). It is also possible to batch-download a large number of sequences from NCBI, by entering their identifiers (accession numbers, GI numbers or GeneIDs) in Batch Entrez (https://www.ncbi.nlm.nih.gov/sites/batchentrez).

### Protein databases

**General information on proteins.** The Universal protein resource (**UniProt**) [17] is currently the main source of information for proteins (**Table 2**). UniProt contains published amino-acid sequences and open-reading-frame translations, with various annotations including structure, classification, biological function, and subcellular localization of the protein. Protein sequences can be downloaded from UniProt in the FASTA format. The resource also provides links to many other public databases. The "retrieve/ID mapping" tool of UniProt (https://www.uniprot.org/id-mapping) facilitates batch downloads of information on a set of proteins using UniProt identifiers. This tool can also be used to convert UniProt identifiers to the identifiers of external databases such as NCBI, GenBank, Ensembl or the Protein Data Bank (PDB). **Gene Ontology** [18] provides a unified annotation system of the molecular function, biological processes, and cellular components of proteins across all species. Information about genes, proteins, and genomes that is acquired from several 'omics technologies is further gathered in the Kyoto Encyclopedia of Genes and Genomes (**KEGG**) [19]. The KEGG database focuses on metabolism, biological pathways, and human diseases. Summaries of the entire human proteome using antibody-based proteomics, transcriptomics, and integration of other omics technologies is gathered in the **Human Protein Atlas** [20]. This atlas displays expression profiles, subcellular localization, tissue and organ distribution, and protein function in human

**Table 2. List of protein databases.**

| Database | Features | Link | References |
|---|---|---|---|
| CATH | Classification of protein domains based on their structure, functionality, and evolution | https://www.cathdb.info/ | [23] |
| FSSP | Classification of protein domains based on their structural similarity | https://archive.ph/20121222235655/http://srs.ebi.ac.uk/srsbin/cgi-bin/wgetz?-page+LibInfo+-id+5Ti2u1RffMj+-lib+FSSP | [24, 25] |
| Gene Ontology | Unified annotation of molecular function, biological processes, and cellular components of proteins | http://geneontology.org/ | [18] |
| Human Protein Atlas | Information on human protein and their link with diseases | https://www.proteinatlas.org/ | [20] |
| InterPro | Classification of proteins domains and functional sites | https://www.ebi.ac.uk/interpro/ | [17] |
| KEGG | Protein function and biological pathways | https://www.genome.jp/kegg/ | [19] |
| PDB | 3D structures of proteins | https://www.rcsb.org/ | [26] |
| Pfam | Information about protein families and domains | http://pfam.xfam.org/ | [27] |
| PHAROS | Centralizes literature for human proteins | https://pharos.nih.gov/ | [21, 22] |
| PRINTS | Protein fingerprints classification database | http://www.bioinf.man.ac.uk/dbbrowser/PRINTS/ | [28] |
| PROSITE | Protein family database | https://prosite.expasy.org/ | [29] |
| SCOP | Classification of protein domains based on their structure, functionality and evolution | https://scop.mrc-lmb.cam.ac.uk/ | [30] |
| SUPERFAMILY | Protein structure and functions | https://supfam.org/ | [31] |
| UniProt | General information on proteins, including sequence, structure, classification, function, subcellular localization and simple homology identification | https://www.uniprot.org/uniprot/ | [17] |

metabolism, as well as information about diseases such as cancer. The **PHAROS** database [20, 21] provides an overview of the literature on human proteins, including their classification, pathways, expression data, and related diseases.

**Protein structure databases.** Protein structures are described by databases such as the **PDB** [26]. The PDB provides three-dimensional (3D) structures of proteins and their interacting ligands established by X-ray crystallography, electron microscopy, or NMR spectroscopy, which can be retrieved as pdb files. The PDB also displays a 3D visualization tool, programs for 3D analyses such as pairwise structure alignment and pairwise symmetry, and cross links to other protein databases. Annotation for protein families based on fingerprints (*i.e.*, conserved 3D motifs specific for a protein family), are gathered in the database **PRINTS** [28]. PRINTS includes a 3D visualization software and search tools for protein sequence homology and pairwise or multiple sequence alignments.

## Protein classification

Proteins are classified into different categories based on structural similarity, functionality and evolutionary relationship (**Table 2**). The 3D structural classification of proteins (**SCOP**) [30] classifies protein domains according to their class, fold, superfamily, and family. Large proteins can have several domains belonging to different categories. The class and fold levels are based on protein structures. Most proteins belong to one of the five structural classes (α, β, α/β, α+β, multi-domain), defined respectively by the presence of α-helices, β-strands, both α-helices and β-strands, segregated α-helices and β-strands, or none of these characteristics. Below this primary level, a protein's secondary structure is reflected in folds. The other levels of protein classification (superfamily and family) are based on evolutionary relationships. Proteins with shared ancestry are classified in the same superfamily, and proteins sharing 30% or more sequence identity are classified in the same family [32]. Similarly, the Class, Architecture,

Topology, Homology (**CATH**) database proposes a five-level classification of protein domains. The first three levels: class, architecture and topology, are based on structural homology. The last two, homologous superfamily and family, are based on sequence, structure and functional specificities, and sequence identity, respectively [23]. The Families of Structurally Similar Proteins (**FSSP**) provides a classification of proteins in the PDB based on a structure comparison algorithm, that calculates a structural similarity score between protein chains. These similarity scores are used to create a classification of protein structures [24, 25].

**SUPERFAMILY** [31] is a database of structurally and functionally annotated proteins. The database Protein families (**Pfam**) [27] classifies protein domains based on multiple sequence alignment. Pfam contains information on protein domain structures and their occurrence in living organisms. The homologues of a protein are listed in Pfam and their sequences can be downloaded in the FASTA format. For any family, Pfam displays a graphical view of all species possessing the protein domain. The domain families and functional sites of proteins is the focus of the **InterPro** database [17]. It combines structure-based and phylogenetic classifications. InterPro uses predictive models called signatures, that are used to infer functions for a sequence in association with the database Gene Ontology [18]. Biological information about the protein family, domains, and functional sites is gathered in the **PROSITE** database [29]. PROSITE also provides tools for identifying distant homology between sequences.

## Homologue research

Studying the evolution of a family of genes or proteins requires the identification of homologues (*i.e.*, genes or protein with shared ancestry). Homologues of a gene or protein can be identified using appropriate tools (**Table 3**). The Basic Local Alignment Search Tool (**BLAST**) [33], maintained at the NCBI, is the most widely used heuristic algorithm for researching sequence homology. Homologues of a gene or protein can be retrieved from the genomes or proteomes of specified taxa with a significance score called E-value (Expect value), which is defined as the number of expected hits of similar quality that can be obtained by chance [34]. A lower E-value corresponds to a higher statistical significance of the match. A BLAST search also provides the identity and similarity of the hits. BLAST can be used for nucleotide (BLASTn) or amino-acid sequences (BLASTp), translated nucleotide to protein (BLASTx) and protein to translated nucleotide (tBLASTn). **UniProt** [17] also provides a tool to identify proteins in the database sharing 50% or 90% identity with any protein, including paralogues and orthologues. **Pfam** [27] and **Ensembl** [7] also include tools to identify homologues of any given gene and retrieve their sequences. Other search tools include **HMMER** [35] of the HH-suite software, **FASTA** [36], **SSAHA** [37] and **BLAT** [38]. Homologues of a given sequence should be compiled into a single FASTA file. FASTA files can be made by retrieving sequences

**Table 3. List of bioinformatic tools for identification of gene and protein homologues.**

| Database | Features | Link | References |
|---|---|---|---|
| BLAST | Protein or DNA homology search in NCBI | https://blast.ncbi.nlm.nih.gov/Blast.cgi | [33] |
| BLAT | Protein or DNA homology search in animal genomes | https://genome.ucsc.edu/cgi-bin/hgBlat | [38] |
| Ensembl | Genome browser of vertebrates, includes tools for identification of homology | https://m.ensembl.org/index.html | [7] |
| FASTA | Protein or DNA homology search and sequence alignment | https://www.ebi.ac.uk/Tools/sss/fasta/ | [36] |
| HMMER | Gene and protein homology search | http://hmmer.org/ | [35] |
| Pfam | Protein families and domains, includes tools for identification of homology | http://pfam.xfam.org/ | [27] |
| SSAHA | DNA sequence search and alignment | https://www.sanger.ac.uk/tool/ssaha/ | [37] |
| UniProt | General information on proteins, includes tools for identification of similarity | https://www.uniprot.org/uniprot/ | [17] |

from the databases and manually adding the sequences one by one. For large datasets, for example the results of an exhaustive search for homologues, it is possible to directly export a large number of sequences in the FASTA format from NCBI or UniProt [17].

## Phylogenetic analysis

Exploring molecular evolution necessitates studying how species, genes, and proteins are related to each other in an evolutionary sense. Phylogenetic relationships can apply to species, genes, or proteins even within the same genome. Reconstructing evolution from molecular data (amino-acid or nucleotide sequences) includes the steps of sequence alignment and trimming, phylogenetic analysis, and study of molecular evolution using a phylogenetic tree. Below, we describe the tools used in these different steps.

## Multiple sequence alignment

Aligning gene or protein sequences consists in inferring homology between bases or amino acids. The sequences are put in every row of a matrix, one after the other, to arrange every homologous base or amino acid. Alignment of the homologous residues necessitates adding gaps, indicated by the symbol "-" and corresponding to insertions or deletions (indels), into the sequences. Sequence alignment methods include the progressive approach and the consistency-based method. The progressive approach aligns progressively from the two closest to the most distant sequences. It is used by **CLUSTALW** [39], **CLUSTAL Omega** [40], **MUSCLE** [41], **PRANK** [37, 55], **KAlign** [44] and **MAFFT** [45]. Consistency-based methods calculate the best multiple sequence alignment (MSA) after different pairwise alignments using information from a third sequence as intermediate [46–48]. They are used by **T-COFFEE** [49], **PROBCONS** [50] and its successor **CONTRAlign** [51], the latter for amino acid sequences only (Table 4). Other approaches include the iterative refinement method, which is also

**Table 4. List of programs for multiple sequence alignment.**

| Software | Features | Link | References |
|---|---|---|---|
| BAli-Phy | Multiple sequence alignment of nucleotide and amino acid sequences and phylogenetic analysis using BI | http://www.bali-phy.org | [59] |
| CLUSTAL Omega* | Speed-oriented multiple sequence alignment for nucleotide or amino acid data | https://www.ebi.ac.uk/Tools/msa/clustalo/ | [40] |
| CLUSTALW* | Multiple sequence alignment for nucleotide or amino acid data | https://www.genome.jp/tools-bin/clustalw | [39] |
| CONTRAlign (ProbCons) | Accuracy-oriented multiple sequence alignment for amino acid data | http://contra.stanford.edu/contralign/ | [50, 51] |
| Kalign* | Speed-oriented multiple sequence alignment for nucleotide or amino acid data | https://www.ebi.ac.uk/Tools/msa/kalign/ | [44] |
| MAFFT* | Multiple sequence alignment for nucleotide or amino acid data | https://mafft.cbrc.jp/alignment/server/ | [45] |
| MUSCLE* | Multiple sequence alignment for nucleotide or amino acid data | https://www.ebi.ac.uk/Tools/msa/muscle/ | [41] |
| PASTA | Speed-oriented multiple sequence alignment for nucleotide or amino acid data, designed for very large datasets | https://bioinformaticshome.com/tools/msa/descriptions/PASTA.html | [60] |
| PRANK/ WebPRANK* | Multiple sequence alignment for nucleotide or amino acid data, should be preferred for close sequences | http://wasabiapp.org/software/prank/ https://www.ebi.ac.uk/goldman-srv/webprank/ | [42, 43] |
| SATé | Software package for multiple sequence alignments and phylogenetic inference | https://phylo.bio.ku.edu/software/sate/sate.html | [62] |
| T-COFFEE* | Multiple sequence alignment of nucleotide and amino acid sequences | http://tcoffee.crg.cat/ | [49] |
| UPP | Speed-oriented multiple sequence alignment of nucleotide and amino acid sequences, designed for very large data sets | https://github.com/smirarab/sepp. | [61] |

(*: include online version).

included in **MAFFT** and **Muscle** [52], the genetic algorithms [53], and methods that use hidden Markov models [54].

Several MSA tools, including **CLUSTALW** [39], **MUSCLE** [41], **MAFFT** [45], **Kalign** [44] and **PRANKS** [37, 55] display inferred MSAs using user interfaces. ClustalW and Muscle are also included in **MEGA** [55]. **PROBCONS** [50], **T-COFFEE** [49] and **MAFFT** [45] are described to have particularly high accuracy but also high execution times [56]. Their use should be restricted to small and intermediate datasets. **CLUSTAL Omega** [40] and **Kalign** [44] are particularly fast, but less accurate [57]. They can be used to analyse datasets of up to 4,000 and 2,000 sequences, respectively [56, 57]. The performances of **MUSCLE** are intermediate [57]. **PRANK** is meant for closely-related sequences [58]. **Bali-Phy** [59] performs a Bayesian co-estimation of alignment, phylogeny, and other parameters and is also argued to be very reliable. **PASTA** [60] and **UPP** [61], which uses a machine-learning technique, are designed for very large datasets. **MAFFT** offers a wide range of methods, which can be accuracy-oriented, such as L-INS-i, G-INS-I and E-INS-i; or speed-oriented, such as FFT-NS-2. The latter can be used for up to 30,000 sequences. Simultaneous Alignment and Tree Estimation (**SATé**) [62] is a software package providing several tools for sequence alignment and phylogenetic analysis.

In practice, finding the accurate MSA can be challenging for several reasons. First, one should keep in mind that the alignment with the best score is not necessarily biologically correct. Computer programs are not based explicitly on the hypothesis of homology between aligned residues [63]. Furthermore, it is difficult to get a good alignment for sequences that have diverged significantly and share low identity. In this case, for protein-coding sequences, amino-acid data should be preferred over nucleotide data, since it is possible to consider the biochemical similarity of amino acids [64]. Alignment programs require defining a gap-opening penalty and a gap-extension penalty, but these values are arbitrary. It is common that different sequences in the alignment do not have the same length, for biological or experimental reasons. It is recommended to keep end-gaps unpenalized [64]. Furthermore, indels are reported to affect the accuracy of MSA programs. It is recommended to use several MSA programs for sequences that contain indels [65]. MAFFT is reported to be the most accurate program in the cases of sequences with non-overlapping deletions and alternatively spliced gene products [65]. Furthermore, single nucleotides, small sequences (e.g., microsatellites) or entire protein domains, can be repeated in a gene or protein sequence. If the number of repeats differs between sequences, one domain of a sequence can be homologous to several domains of another sequence. It is recommended to excise the repeated domains [64].

## Alignment trimming

Once the alignment is completed, it is necessary to select the positions and regions that will be used for the phylogenetic inference. Poorly aligned positions and highly variable regions are not phylogenetically informative, because these positions might not be homologous or subject to saturation. They should be excluded prior to the phylogenetic analysis to maximize the phylogenetic signal of the alignment [66]. A minimum reporting standard has been developed to quantify the alignment completeness, and implemented in **AliStat** [67]. Phylogenetically informative regions of the alignment can be selected using appropriate tools, such as **Guidance 2** [68], **GBlocks** [69], **trimAl** [70], **BMGE** [71] and **Noisy** [72] (**Table 5**).

## Assessing phylogenetic assumptions

Phylogenetic methods rely on simple assumptions about the evolutionary processes, stating, for example, that all sites in the alignment evolved under the same tree (treelikeness), that

**Table 5. List of programs for sequence alignment trimming.**

| Software | Features | Link | References |
|---|---|---|---|
| AliStat | Quantification of alignment completeness for alignment refinement | https://github.com/thomaskf/AliStat | [67] |
| BMGE | Selection of informative regions on multiple sequence alignments | https://gitlab.pasteur.fr/GIPhy/BMGE | [71] |
| GBlocks | Selection of informative regions on multiple sequence alignments | http://molevol.cmima.csic.es/castresana/Gblocks.html | |
| Guidance 2* | Selection of informative regions on multiple sequence alignments | http://guidance.tau.ac.il/ | [68] |
| Noisy | Selection of informative regions on multiple sequence alignments | http://www.bioinf.uni-leipzig.de/Software/noisy/ | [72] |
| trimAl | Selection of informative regions on multiple sequence alignments | http://trimal.cgenomics.org/ | [70] |

(*: tools that include online version).

mutation rates have remained constant over time (time-homogeneity), and that substitutions are reversible and, therefore, also stationary (for details on these assumptions, see [73]). If the phylogenetic data violate these assumptions, the phylogeny and evolutionary analyses may become biased [74–76]. Once the alignment is performed and the sites have been selected for phylogenetic inference, it is recommended to assess those phylogenetic assumptions when possible [1]. Statistical methods allowing users to test stationarity and homogeneity of the evolutionary processes (along diverging lineages), and treelikeness, have been developed and included in IQ-TREE and IQ-TREE2 [77, 78]. Homo2.1 (https://github.com/lsjermiin/Homo.v2.1) [79] is designed for the analysis of compositional heterogeneity in sequence alignments. It is also possible to use the R package MOTMOT [80].

## Phylogenetic protocol selection

The choice of a phylogenetic method can be challenging. The appropriate phylogenetic method depends on the phylogenetic assumptions of each method. Inferring the correct phylogenetic tree requires that the data do not violate the assumptions of the method. Most phylogenetic methods assume that the sequences have evolved under time-reversible Markovian conditions (*i.e.*, the nucleotides or amino-acids have evolved independently of time and their past history). Most model selectors consider only time-reversible Markovian models. However, if the data have evolved under more complex, non-time reversible Markovian conditions, identifying the sequence evolution model that fits the data and reconstructing the phylogenetic tree may be complex, since phylogenetic methods for such data are lacking [1]. A model of sequence evolution selected as the best fit does not necessarily imply that it adequately describes the data. Poorly fitting models are inadequate approximations of the evolutionary processes and can lead to errors. In this case, it is recommended to test the goodness of fit between the phylogenetic tree, the substitution model, and the data (see paragraph Test of goodness of Fit).

## Selection of the optimal model of sequence evolution

Probabilistic and distance methods require selection of the model of molecular evolution that best describes the data. Several models of nucleotide or amino-acid substitution exist [81]. The nucleotide substitution models differ in the number of parameters considered, like mutation rates and base frequencies [82] (for a review see [83]). The main nucleotide substitution models are, from the simplest to the most complex: JC69 [84], K80 [85], F81 [86], HKY85 [87], TN93 [88] and GTR [89]. The main amino acid substitution models include JTT [90], WAG [91], LG [92] and Dayhoff [93]. Models of codon evolution also exist [94].

These models can be associated with models of substitution rate heterogeneity across sites. Mutation rates and selective pressure may vary among sites, due to different roles in the

**Table 6. List of programs for molecular evolution model selection.**

| Software | Features | Link | References |
|---|---|---|---|
| ModelFinder | Fast model selection method with a model of rate heterogeneity between sites (nucleotides, amino acids, codons) | Implemented in IQ-Tree | [101] |
| ModelTest / jModelTest | Nucleotide substitution model selection | http://evomics.org/resources/software/molecular-evolution-software/modeltest/ | [99] |
| PartitionFinder 2 | Molecular evolution model selection | http://www.robertlanfear.com/partitionfinder/ | [102] |
| ProtTest | Aminoacid substitution model selection (nucleotides, amino acids) | https://github.com/ddarriba/prottest3 | [100] |
| SMS | Nucleotide or aminoacid model selection included in PhyML (nucleotides, amino acids) | http://www.atgc-montpellier.fr/sms/ | [103] |

structure and function of the gene or protein. The most common rate heterogeneity across sites models are the Gamma distribution (G) and the proportion of invariant nucleotide or amino acid sites (I). Every substitution model can be associated with G, I or both. The FreeRate model (R), a more complex model of rate heterogeneity [95], is included in ModelFinder, PhyML and IQ-TREE. More recently, the GHOST model for alignments with variation in mutation rate was introduced and implemented in IQ-TREE [96].

The likelihood of the different models should be computed by appropriate software (Table 6). For every model of sequence evolution (i.e., a combination of a substitution model and a rate-heterogeneity across sites model), these tools calculate the Bayesian information criterion (**BIC**) [97] and the Akaike information criterion (**AIC**) [98] from the log-likelihood scores. A model with lower BIC or AIC is considered more accurate. The model minimizing BIC or AIC (*i.e.*, with the lowest score) should be selected. **ModelTest** and **jModelTest** [99, 100] estimate the likelihood for phylogenetic trees based on nucleotide sequences and **ProtTest** [100] for amino acid sequences. **ModelFinder** [101] is a model selection method, for alignments of nucleotides, codons or amino acids, implemented in IQ-TREE [77, 78]. **PartitionFinder 2** [102] can be used with nucleotide and amino acid data. Model selectors are also included in programs such as MEGA [55] and PhyML (**SMS**) [103].

## Phylogenetic analysis

A phylogenetic tree is a graphical illustration of the evolutionary relationships between taxa, genes, or proteins. For comprehensive reviews, see [104, 105]. Phylogenetic trees may consider the topology and the branch lengths (phylograms) or just the topology (cladograms). Several tree-building methods exist. Distance methods create a matrix of molecular distances, defined by the numbers and types of differences between the sequences, and they use this matrix to reconstruct the phylogenetic tree. Character-based methods compare all sequences at the same time, site by site. They include Maximum Parsimony (**MP**) and the probabilistic methods: Maximum Likelihood (**ML**) [106] and Bayesian Inference (**BI**) [107–109]. Maximum parsimony is a classical and simple method, now rarely used with molecular data, that calculates the minimum number of evolutionary steps, including nucleotide insertions, deletions or substitutions, between species. The main weakness of this method is that it ignores hidden mutations and does not consider branch lengths. This can lead to incorrect clustering of unrelated taxa, a phenomenon known as long branch attraction (for a review, see [110]). Probabilistic methods are the most recent and today the most widely used methods for phylogenetic inference. They are more relevant for molecular phylogenetics because they use specified models of molecular evolution and rely on likelihood calculations, but their execution time is longer.

**Phylogenetic analysis using distance methods.** Distance methods include the Unweighted Pair Group Method with Arithmetic Mean (**UPGMA**) [111], Neighbor Joining

**Table 7. List of programs for phylogenetic analysis using distance methods, maximum parsimony, maximum likelihood and Bayesian inference.**

| Software | Features | Link | References |
|---|---|---|---|
| APE | R-written package for molecular phylogenetics using distance-based methods | http://ape-package.ird.fr | [118] |
| BAli-Phy | Phylogenetic inference using BI | http://www.bali-phy.org | [59] |
| BayesTraits | Phylogenetic inference and other evolutionary analyses using BI | http://www.evolution.reading.ac.uk/BayesTraitsV4.0.0/BayesTraitsV4.0.0.html | [136] |
| FastMe* | Phylogenetic inference using distance methods | http ://www.atgc-montpellier.fr/fastme/ | [114] |
| GARLI | Phylogenetic inference using ML | http ://evomics.org/resources/software/molecular-evolution-software/garli/ | [137] |
| HYPHY* | Phylogenetic inference using ML and distance methods | https ://www.hyphy.org/ | [128] |
| IQ-TREE* | Phylogenetic inference using ML, including model selection and a very fast bootstrapping method | http ://www.iqtree.org/ | [77, 78] |
| MEGA | Sequence alignment, model selection, phylogenetic analysis using distance methods, MP and ML, and other evolutionary analyses | https://www.megasoftware.net/ | [55] |
| MrBayes | Phylogenetic inference using BI and diverse evolutionary analyses, including ancestral states reconstruction and time calibration | http://nbisweden.github.io/MrBayes/ | [133] |
| PAML | phylogenetic inference using ML, estimation of selection strength, ancestral states reconstruction and other evolutionary analyses | http://abacus.gene.ucl.ac.uk/software/paml.html | [127] |
| PAUP | Phylogenetic inference using MP and ML | http://paup.phylosolutions.com/ | [119] |
| PHYLIP | Phylogenetic inference using MP, distance methods and ML | https ://evolution.genetics.washington.edu/phylip.html | [121] |
| PhyloBayes | Phylogenetic inference with protein data using BI using a specific probabilistic model | http ://www.atgc-montpellier.fr/phylobayes/ | [134] |
| PhyML* | Phylogenetic inference using ML, ancestral states reconstruction and various evolutionary analyses | http://atgc.lirmm.fr/phyml/ | [125] |
| PyCogent | Phylogenetic inference, tree drawing, various evolutionary analyses, including partition models and ancestral states reconstruction | https ://github.com/pycogent/pycogent | [115] |
| RAxML* | Phylogenetic inference using ML | https ://cme.h-its.org/exelixis/web/software/raxml/ | [126] |
| SeaView | Sequence alignment and phylogenetic inference using MP, NJ and ML | http://doua.prabi.fr/software/seaview | [124] |
| SplitsTree | Phylogenetic inference for unrooted trees and phylogenetic networks | https ://uni-tuebingen.de/fakultaeten/mathematisch-naturwissenschaftliche-fakultaet/fachbereiche/informatik/lehrstuehle/algorithms-in-bioinformatics/software/splitstree/ | [117] |

(**NJ**) [112], and Minimum Evolution (**ME**) [113] tree-inference methods. **FastME** [114] is designed for phylogenetic inference using diverse distance methods with nucleotide or amino-acid data or distance matrices (**Table 7**). **PyCogent** [115] is a software library for genomic biology, allowing for phylogenetic analysis and a large number of evolutionary, statistical and genomic analyses, including partition models, as well as graphical display and annotation of phylogenetic trees. **SplitsTree** [116, 117] is used for inference of unrooted phylogenetic trees and phylogenetic networks from sequence alignments or distance matrices. **APE** (Analysis of Phylogeny and Evolution) [118] is a package written in the R language that provides a wide range of evolutionary analyses, including calculating genetic distances and computing phylo-genetic trees using distance-based methods. **PAUP** [119], **MEGA** [55], **FastTree** [120] or **PHYLIP** [121], can also be used for distance-based methods, as well as ML, and/or MP [122, 123].

**Phylogenetic analysis using maximum likelihood.** Maximum-likelihood methods calcu-late the likelihood of observing the data under different explicit models of molecular evolution. Maximum likelihood aims to identify the best fit model by exploring multiple combinations of trees and model parameters. Programs for ML phylogenetic analysis include **MEGA** [55],

**SeaView** [124], **PhyML** [125], **RAxML** [126], **FastTree** [120], **PAML** [127], **PAUP** [119], **IQ-TREE** [77, 78], **HYPHY** [128], **PHYLIP** [121] and **GARLI** [129] (**Table 7**). All of them can be used with nucleotide or amino-acid data. **MEGA** [55] and **SeaView** [124] are known to be very user-friendly. They include sequence alignment tools and tree manipulators. **PhyML** [125] is reported as being accurate, easy to use and, like **PAUP** and **MEGA** [55], includes many common models of substitution. **RAxML** [126] and particularly **FastTree** [120] are fast and well suited for large datasets (up to 1 million sequences with FastTree). In addition to assuming Gamma-distributed rate-heterogeneity across sites and the proportion of invariant sites, they include CAT, a specific model of rate heterogeneity [130]. **IQ-TREE** [77, 78], which includes ModelFinder [101] and the very fast bootstrapping method UFBoot2 [131], is reported to be both fast and accurate [132].

## Phylogenetic analysis using Bayesian Inference

The most recently-developed method for phylogenetic reconstruction uses Bayesian Inference (BI). This method calculates the posterior probability of the tree and model of sequence evolution, given the data. The main software used for BI-based phylogenetics is **MrBayes** [133]. It uses the Markov Chain Monte Carlo (MCMC) algorithm (**Table 7**). **PhyloBayes** [134, 135] is a Bayesian MCMC sampler for phylogenetic reconstruction with protein data using a specific probabilistic model. It is well adapted for large datasets and phylogenomics. **Bali-Phy** [59] can also be used for phylogenetic analysis using BI.

## Test of reliability of the inferred tree

It is recommended to estimate the reliability of the clades of the inferred phylogenetic tree. Most programs of phylogenetic analysis use the non-parametric bootstrapping method [138]. Bootstrapping is a resampling technique used to assess the repeatability of the clade, and estimate how consistently it is supported by the data [139]. The sites in the alignment (nucleotides or amino acids) are randomly resampled with replacement and a new phylogeny is calculated for each replicate [138]. A bootstrap value, corresponding to the proportion of replicate phylogenies that recovered the clade, is calculated for every internal branch. A bootstrap value of 100% means that the branch is supported by all resampled datasets, while low values mean that only few of these datasets support the branch. Bootstrap values depend on both data and the method used. Users should keep in mind that bootstrapping gives a measure of the consistency of the estimate, but it is not a measure of the accuracy of the tree [140]. The number of replicates that are necessary to obtain a good accuracy of the bootstrap depends on the bootstrap value. For example, for a 1% confidence interval on a bootstrap value of 95, 2,000 replicates are necessary [139].

Since bootstrapping can be time consuming, fast approximation methods for phylogenetic bootstrap, UFBoot and UFBoot2, have been developed and implemented in IQ-TREE [131, 141, 142]. They are also less biased than other non-parametric bootstrapping methods and robust against moderate model violations. While other methods tend to underestimate the probabilities of the clade of being correct, the values from UFBoot and UFBoot2 truly reflect this probability, simplifying the interpretation of bootstrap values [141].

The approximate likelihood ratio test (aLRT), implemented in PhyML [143], is an alternative to the non-parametric bootstrap. Bayesian inference methods use posterior probabilities (PP) to measure branch support. It is also possible to compare the topology of different trees. In ML-based phylogenetic analysis, the Shimodaira-Hasegawa (SH) test and its improved version, Approximately unbiased (AU) [144], have been designed to evaluate alternative phylogenetic hypotheses, and test if a tree is better supported than another one. This test can be used with PAUP, PhyML, FastTree and IQ-TREE.

## Tree rooting

The root of a phylogenetic tree is the hypothetical last common ancestor of all the sequences present in the tree. Depending on the question asked, phylogenetic trees can be unrooted or rooted. The latter corresponds to the identification of ancestral and derived states, aiming at studying the direction of the evolution of the sequences [69]. Diverse methods have been developed to root phylogenetic trees. The most common consists in including outgroups (*i.e.*, sequences that are closely related to the ingroup of interest) in the analysis. Typically, two outgroups are selected, one being more closely related to the ingroup than the other, allowing for a proper identification of the states of characters. Correctly rooting a phylogeny can be challenging, for example in the case of rapid evolutionary radiations. The outgroups can be subject to long-branch artifacts and tend to cluster with the longest branches of the tree [145]. A study suggests reconstructing the trees with and without outgroups. When the outgroup affects the topology, the tree with no outgroups should be preferred [146]. When outgroups are not included, alternative methods can be used. For example, midpoint rooting places the root at the mid-point between the most dissimilar sequences in the tree, and molecular clock rooting assumes that evolution speed is constant between the sequences [69].

## Test of goodness-of-fit

The inferred optimal model of sequence evolution used for the phylogenetic analysis can be inadequate. Once a phylogenetic tree has been inferred, it is recommended to test the goodness-of-fit (*i.e.*, the adequacy) between the tree, the model, and the data [1]. A good fit means that the tree and the model of sequence evolution provide a good explanation of the data but does not indicate if the tree is correct or not. The goodness of fit can be tested using a parametric bootstrap [147], a method that consists in simulating sequence evolution to generate pseudo-data, using the optimal tree and the optimal model as an input. Sequence generating programs, such as **SeqGen** (https://github.com/rambaut/Seq-Gen/releases/tag/1.3.4, [148]) can be used. The goodness-of-fit is calculated from the difference between the unconstrained and constrained (*i.e.*, assuming the optimal tree and model) log-likelihoods of the real data and the pseudo-data. If the fit is poor, it is recommended to check the alignment and the selected set of sites, and the sequence evolution model (feedback loops on **Fig 1**). The adequacy of the data can be tested using the frequentist Goldman-Cox (GC) test, which can be performed with PAUP [149]. Most Bayesian phylogenetic programs employ the posterior predictive (PP) test [150].

**Table 8. List of tools for graphical visualization and annotation of phylogenetic trees.**

| Software | Features | Link | References |
|---|---|---|---|
| ETE Toolkit | Visualization and analysis of phylogenetic trees | http://etetoolkit.org/ | [152] |
| FigTree | Graphic software for phylogenetic trees | https://github.com/rambaut/figtree/releases | [151] |
| ITOL* | Visualization and annotation of phylogenetic trees | https://itol.embl.de/ | [153] |
| MEGA | Sequence alignment, model selection, phylogenetic analysis, includes tree visualization and annotation tools | https://www.megasoftware.net/ | [55] |
| SeaView | Sequence alignment and phylogenetic inference, includes tree visualization and annotation tools | http://doua.prabi.fr/software/seaview | [124] |

(*: includes online version).

## Visualization tools

Once the phylogenetic tree has been computed, it can be visualized using graphical software such as **FigTree** [151], **ETE Toolkit** [152] or **ITOL** [153] (**Table 8**). **MEGA** [55] and **SeaView** [124] also include visualization tools. Using different sets of options, several types of phylogenetic trees can be drawn (rooted or not, cladogram or phylogram), and branch support values (bootstrap values or posterior probabilities) can be displayed.

## Integrative services for phylogenetic workflows

Packages for phylogenetic analysis can facilitate phylogenetic inference, analysis and other evolutionary studies (**Table 9**). A complete series of libraries for bioinformatics including tools for sequence alignment, phylogenetic analysis, study of molecular evolution and population genetics is available through the **Bio++** suite [154] and **HyPhy** [128]. The Cyberinfrastructure for phylogenetic research (**CIPRES science gateway**) [155] is a public resource for phylogenetic analysis that includes many tools and software for sequence alignment, model selection, and phylogenetic inference. Other packages include **NGPhylogeny** [156] and **Phylemon** [157]. **Geneious** is a platform for DNA, RNA and protein studies that provides tools for NGS assembly, sequence alignment, phylogenetic analysis using NJ, UPGMA, ML and BI, 3-D structures study, and SNP analysis [158].

## Study of molecular evolution

Inferring phylogenetic relationships allows users to study many aspects of molecular evolution. Here, we propose a non-exhaustive list of studies that can be carried out using phylogenetic trees and the above mentioned bioinformatic tools.

## Reconstitution of ancestral states

Retracing the functional evolution of genes, proteins, or biological traits often requires the reconstitution of ancestral states. Ancestral states can be inferred from a phylogenetic tree using MP, ML, or BI. To infer ancestral states also requires the aligned sequences and, when using probabilistic and distance methods, the model of sequence evolution that has been used for the phylogenetic analysis. For ML-based reconstructions, **MEGA** [55], **PAML** [127], **IQ-TREE** and **IQ-TREE 2** [77, 78], **HyPhy** [128], **Bio++** [154], and **Mesquite** [159] can be used (**Table 10**). **BEAST** [160], **MrBayes** [133], and **BayesTraits** [136] use BI. **RASP** (Reconstruct Ancestral state in Phylogenies) [161] can be used for both ML- and BI-based ancestral states reconstruction. **PyCogent** [115] provides a large number of evolutionary analyses, including ancestral states reconstruction.

**Table 9. List of packages for phylogenetic analysis and evolutionary studies.**

| Software | Features | Link | References |
|---|---|---|---|
| Bio++ | Software package for phylogenetic analysis | https://github.com/BioPP | [154] |
| CIPRES | Resource for evolutionary studies, that includes programs for sequence alignment, model selection, and phylogenetic inference | https://www.phylo.org/ | [155] |
| HyPhy | Software package for phylogenetic analysis | https://www.hyphy.org/ | [128] |
| Geneious | Platform for NGS data assembly and diverse evolutionary studies | https://www.geneious.com/ | [158] |
| NGPhylogeny | Workflow that integrates numerous methods and tools for phylogenetic analysis | https://ngphylogeny.fr | [156] |
| Phylemon2 | Web-tools for molecular evolution, phylogenetics, phylogenomics and hypotheses testing. | http://phylemon.bioinfo.cipf.es/index.html | [157] |

**Table 10. List of programs and databases for diverse evolutionary analyses in complement to phylogenetic analysis.**

| Software | Features | Link | References |
|---|---|---|---|
| Arlequin | Population genetics analyses | http://cmpg.unibe.ch/software/arlequin35/ | [162] |
| BayesTraits | Evolutionary analyses using Bayesian inference | http://www.evolution.reading.ac.uk/BayesTraitsV4.0.0/BayesTraitsV4.0.0.html | [136] |
| BEAST | Diverse evolutionary analyses using BI, including time-calibration of phylogenetic trees | http://www.beast.community | [160] |
| CAFE | Gene family evolution | https://github.com/hahnlab/CAFE5 | [163] |
| CoGE | Comparative genomics analyses | https://genomevolution.org/coge/ | [164] |
| Copycat | Coevolution studies | http://www.cophylogenetics.com/ | [165] |
| CoRe-PA | Coevolution studies | http://pacosy.informatik.uni-leipzig.de/49-1-CoRe-PA.html | [166] |
| DNAsp | Analysis of DNA polymorphism | http://www.ub.edu/dnasp/ | [167] |
| Genepop | Population genetics analyses | https://genepop.curtin.edu.au/ | [168] |
| HGT-Finder | Horizontal gene transfer finding | https://github.com/yinlabniu/HGT-Finder | [169] |
| IQ-TREE, IQ-TREE2 | Ancestral states reconstruction | http ://www.iqtree.org/ | [77, 78] |
| Jane | Coevolution studies | https://www.cs.hmc.edu/~hadas/jane/ | [170] |
| LSD | Time-calibration of phylogenetic trees | http ://www.atgc-montpellier.fr/LSD/ | [178] |
| Mesquite | Comparative analyses and statistics | http://www.mesquiteproject.org/ | [159] |
| Ohnologs | Database of vertebrate ohnologues, resulting from whole genome duplications | http://ohnologs.curie.fr/ | [171] |
| PyCogent | Numerous evolutionary analyses, including partition models and phylogenetic analysis, tree drawing and ancestral states reconstruction | https ://github.com/pycogent/pycogent | [115] |
| RASP | Ancestral states reconstruction | http://mnh.scu.edu.cn/soft/blog/RASP/index.html | [161] |
| SNiplay | SNP detection and other population genetics analyses | https://sniplay.southgreen.fr/cgi-bin/home.cgi | [172] |
| TimeTree | Trime-calibration of phylogenetic trees | http://www.timetree.org/ | [173] |
| TreeMap | Software for studying co-evolution | https://sites.google.com/site/cophylogeny/treemap | [174] |

## Measure of selection strength

The type and strength of selection on protein coding genes may be of interest. It is calculated by evaluating the ratio of the number of non-synonymous substitutions (substitutions changing the protein sequence) per non-synonymous site (dN), and the number of synonymous substitutions (substitutions with no effect on the protein sequence due to the redundancy of the genetic code) per synonymous site (dS). If dN/dS > 1, then the non-synonymous substitutions are higher than expected and the gene is under positive selection. If dN/dS<1, the gene is under purifying selection and if dN/dS = 1, the selection is neutral. It is recommended not to use the dN/dS ratio for closely related species [175]. The ratio can be calculated using **PAML** [127], **MEGA** [55], **Bio++** [154] and **HyPhy** [128] (**Table 10**).

## Time-calibration of phylogenetic trees

Time calibration of phylogenetic trees consists in estimating divergence times, using events with a known age, such as fossil and other geological data (that can only give minimal ages) as calibration points. Alternatively, mutation rates can be used to calculate the divergence time between two sequences. However, since the mutation rate can vary a lot during long-term evolution and differ between taxonomic groups, using mutation rates should be avoided for distantly related species [176]. The estimated divergence times between species is summarized in **TimeTree** [173] (**Table 10**). It is noteworthy that divergence times estimates from the literature, based on calibration points from fossil data and molecular clocks, are prone to error and illusory precision [177]. TimeTree can be used with MEGA [55] to calibrate a phylogeny. The

**BEAST** package [160] uses BI to estimate mutation rates and calibrate phylogenies. **LSD** [178], recent versions of **PhyloBayes** [134] and **APE** [118] can also be used for molecular dating of evolutionary events.

## Study of host/parasite co-evolution

Co-evolution refers to the genetic or morphological changes (or both) between different species in interaction. It is widely used in evolutionary ecology and parasitology to study the evolution of hosts and parasites. Co-evolutionary events include co-speciation, host change, duplication, and loss of interaction. The evolution of the parasite is partly driven by the evolution of the host, which is considered independent from the evolution of the parasite [179]. The co-evolutionary history can be presented as a co-phylogeny with the two entities. Some programs for studying co-evolution, including **Jane** [170], **CoRe-PA** [166] and **TreeMap** [174] (**Table 10**), are based on the hypothesis that the evolution of the parasite is driven by the evolution of the host. Others, such as **Copycat** [165], reconcile the two phylogenies under the hypothesis that the situation is symmetric and evaluate the significance of co-evolution under a statistical framework. Co-evolution of genes or proteins can also be studied using these tools.

## Phylogenetic comparative analysis

Evolutionary biology often employs the so-called phylogenetic comparative methods to study the adaptive significance of biological traits. These methods aim at identifying biological characters, in terms of morphology, physiology or ecology, that result from a shared ancestry. Comparative analysis uses a correlative approach between traits, taking into account the phylogenetic constraints [180]. Comparative analyses can be performed for quantitative or qualitative variables. Suitable programs include **Mesquite** [159] and **BayesTraits** [136] (**Table 10**).

## Genome evolution

Phylogenetic trees, in complement with genomics tools and databases, can be used to study genome evolution, and identify evolutionary events such as mutations, insertions, deletions, gene or genome duplications, genome re-organization, chromosomal rearrangements, polyploidization events or genetic exchanges. Molecular databases, such as **Ensembl** [7] and **GenBank** [4] (**Fig 1**, **Table 1**), can be used to study genome evolution. **Ohnologs** [171] summarizes the whole genome duplication events during the evolution of vertebrates. This database can be used to interpret the duplication events and identify paralogues resulting from a whole genome duplication. Horizontal gene transfer, *i.e.*, the gene exchanges between different organisms, can be estimated using **HGT-Finder** [169] (**Table 10**). **CoGe** [164] provides many tools for comparative genomic research, including BLAST [33] and tools for studying synteny, genomic inversions or horizontal gene transfers. Computational analysis of gene family evolution (**CAFE**) [163] is a program for studying the evolution of gene family sizes. It can be used to calculate the birth and death rates of gene families over phylogenies.

## Population genetics

Genetic diversity can also be explored at the population level by analyzing polymorphism between members of the same species. Population geneticists often study allele diversity within a population, including single nucleotide polymorphisms (SNP), indels, microsatellites or transposable elements. Mathematical models have been developed to describe polymorphism. For instance, nucleotide diversity ($\pi$) measures the degree of polymorphism in a population, based on the average number of SNPs per site [181]. The fixation index ($F_{ST}$) is a statistic of

**Table 11. List of programs for protein structure analyses.**

| Software | Features | Link | References |
|---|---|---|---|
| Alpha fold | Protein structure prediction from amino-acid sequence | https://alphafold.ebi.ac.uk/ | [188] |
| FoRSA | Protein structure prediction from amino-acid sequence | http://www.bo-protscience.fr/forsa/ | [189] |
| HHPred | Protein structure prediction from amino-acid sequence | https://toolkit.tuebingen.mpg.de/tools/hhpred | [187] |
| I-TASSER | Protein structure prediction from amino-acid sequence | https://zhanglab.dcmb.med.umich.edu/I-TASSER/ | [186] |
| PyMOL | 3D visualization of molecules | http://www.mesquiteproject.org/ https://pymol.org/2/ | [185] |

genetic distance between populations based on their allelic composition using multiple alleles [182]. Linkage disequilibrium measures the association between alleles at different loci in a population. Several programs are suitable for population genetics studies; for a full review, see [183]. **Arlequin** [162], **SNiPlay** [172], **DNAsp** [184] and **GENEPOP** [168] can be used to compute statistics describing genetic diversity in populations, as well as the R-written package **APE** [118] (**Table 10**). Arlequin and GENEPOP are also relevant for inferring the strength of genetic drift and selection. The **Bio++** suite [154] and **HyPhy** [128] also include tools for population genetics analyses.

## Protein structure study

The study of protein functional evolution can require bioinformatic tools for protein structural analyses (**Table 11**). 3D structure comparisons can be performed using **PyMOL** [185]. Structure alignments can be realized and the mean distance in ångström between homologous residues can be calculated with this program. **I-TASSER** [186], **HHPred** of the HH suite [187] and **Alpha fold** [188] can be used to predict the 3D structure of proteins from their amino-acid sequences. **FoRSA** [189] is able to identify a protein fold from its amino-acid sequence or a protein sequence in the proteome of a species from a crystal structure.

## Two test-cases of evolutionary analyses of proteins

Following the roadmap for evolutionary analyses of proteins that is presented above (**Fig 1**), we now demonstrate how to track the evolution of the p53 family and human cyclins and CDKs.

## Reconstructing the evolutionary history of the p53 family

*TP53* is a transcription factor regulating genes involved in DNA repair and cell cycle control, inducing growth arrest or apoptosis depending on the physiological conditions and cell type. *TP53* has been extensively studied for its role in development and cancer. Two paralogues of p53 are identified in vertebrate genomes: p63 and p73. Here, we propose a simple bioinformatic study to reconstruct the evolutionary history of the proteins p53, p63 and p73, to illustrate our roadmap. We investigate how the paralogues of different animal species are related to each other, when they appeared and diverged, and when they evolved new protein domains. We describe step-by-step the methods and tools used, from the selection of sequences to the phylogenetic inference and reconstruction of the evolutionary history of the TP53 family, using data from reference [190].

**1. State of the art and protein classification of p53.** According to UniProt [17], the human p53 (cellular tumor antigen p53, hereafter named HsTP53) is located in the nucleus, the endoplasmic reticulum, the cytoskeleton and the mitochondrion. HsTP53 is labelled as P04637 in UniProt [17] (https://www.uniprot.org/uniprot/P04637, accessed September 06, 2021) where the full sequence can be downloaded in the FASTA format. HsTP53 contains 393

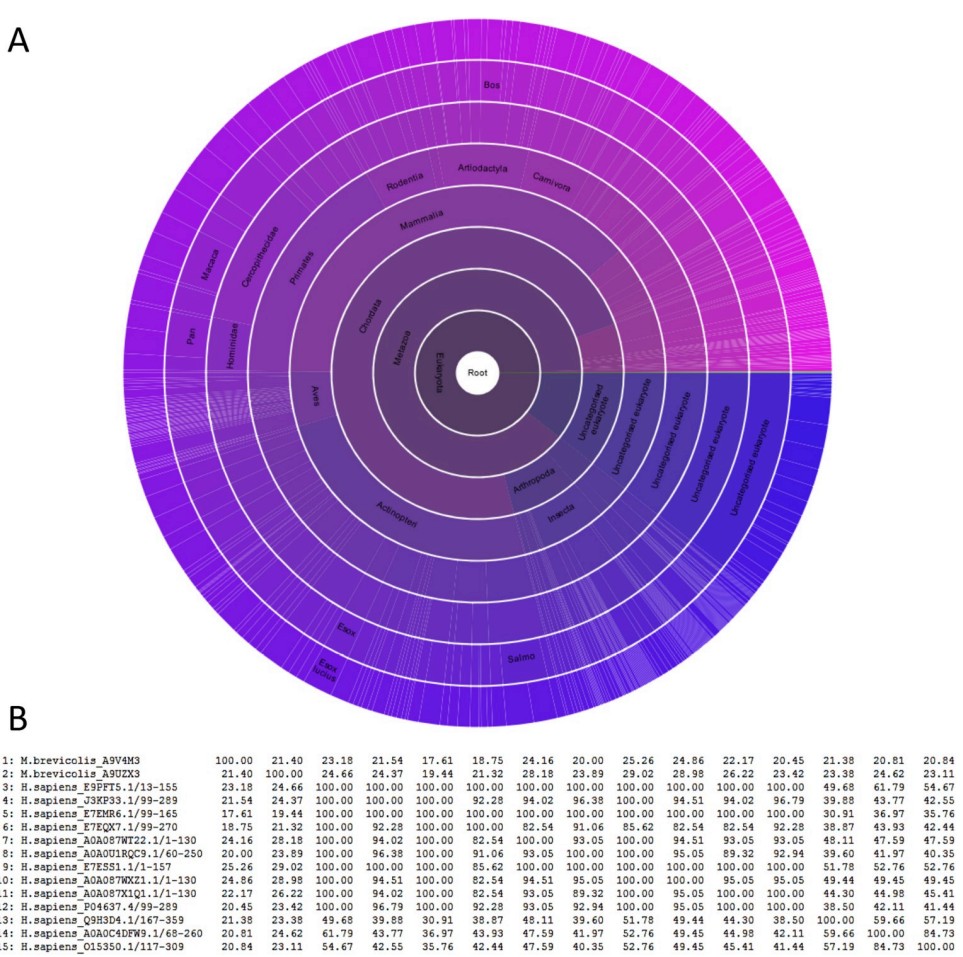

**A**

**B**

| | | | | | | | | | | | | | | | |
|---|---|---|---|---|---|---|---|---|---|---|---|---|---|---|---|
| 1: M.brevicolis_A9V4M3 | 100.00 | 21.40 | 23.18 | 21.54 | 17.61 | 18.75 | 24.16 | 20.00 | 25.26 | 24.86 | 22.17 | 20.45 | 21.38 | 20.81 | 20.84 |
| 2: M.brevicolis_A9UZX3 | 21.40 | 100.00 | 24.66 | 24.37 | 19.44 | 21.32 | 28.18 | 23.89 | 29.02 | 28.98 | 26.22 | 23.42 | 23.38 | 24.62 | 23.11 |
| 3: H.sapiens_E9PFT5.1/13-155 | 23.18 | 24.66 | 100.00 | 100.00 | 100.00 | 100.00 | 100.00 | 100.00 | 100.00 | 100.00 | 100.00 | 100.00 | 49.68 | 61.79 | 54.67 |
| 4: H.sapiens_J3KP33.1/99-289 | 21.54 | 24.37 | 100.00 | 100.00 | 100.00 | 92.28 | 94.02 | 96.38 | 100.00 | 94.51 | 94.02 | 96.79 | 39.88 | 43.77 | 42.55 |
| 5: H.sapiens_E7EMR6.1/99-165 | 17.61 | 19.44 | 100.00 | 100.00 | 100.00 | 100.00 | 100.00 | 100.00 | 100.00 | 100.00 | 100.00 | 100.00 | 30.91 | 36.97 | 35.76 |
| 6: H.sapiens_E7EQX7.1/99-270 | 18.75 | 21.32 | 100.00 | 92.28 | 100.00 | 100.00 | 82.54 | 91.06 | 85.62 | 82.54 | 82.54 | 92.28 | 38.87 | 43.93 | 42.44 |
| 7: H.sapiens_A0A087WT22.1/1-130 | 24.16 | 28.18 | 100.00 | 94.02 | 100.00 | 82.54 | 100.00 | 93.05 | 100.00 | 94.51 | 93.05 | 93.05 | 48.11 | 47.59 | 47.59 |
| 8: H.sapiens_A0A0U1RQC9.1/60-250 | 20.00 | 23.89 | 100.00 | 96.38 | 100.00 | 91.06 | 93.05 | 100.00 | 100.00 | 95.05 | 89.32 | 92.94 | 39.60 | 41.97 | 40.35 |
| 9: H.sapiens_E7ESS1.1/1-157 | 25.26 | 29.02 | 100.00 | 100.00 | 100.00 | 85.62 | 100.00 | 100.00 | 100.00 | 100.00 | 100.00 | 100.00 | 51.78 | 52.76 | 52.76 |
| 10: H.sapiens_A0A087WXZ1.1/1-130 | 24.86 | 28.98 | 100.00 | 94.51 | 100.00 | 82.54 | 94.51 | 95.05 | 100.00 | 100.00 | 95.05 | 95.05 | 49.44 | 49.45 | 49.45 |
| 11: H.sapiens_A0A087X1Q1.1/1-130 | 22.17 | 26.22 | 100.00 | 94.02 | 100.00 | 82.54 | 93.05 | 89.32 | 100.00 | 95.05 | 100.00 | 100.00 | 44.30 | 44.98 | 45.41 |
| 12: H.sapiens_P04637.4/99-289 | 20.45 | 23.42 | 100.00 | 96.79 | 100.00 | 92.28 | 93.05 | 92.94 | 100.00 | 95.05 | 100.00 | 100.00 | 38.50 | 42.11 | 41.44 |
| 13: H.sapiens_Q9H3D4.1/167-359 | 21.38 | 23.38 | 49.68 | 39.88 | 30.91 | 38.87 | 48.11 | 39.60 | 51.78 | 49.44 | 44.30 | 38.50 | 100.00 | 59.66 | 57.19 |
| 14: H.sapiens_A0A0C4DFW9.1/68-260 | 20.81 | 24.62 | 61.79 | 43.77 | 36.97 | 43.93 | 47.59 | 41.97 | 52.76 | 49.45 | 44.98 | 42.11 | 59.66 | 100.00 | 84.73 |
| 15: H.sapiens_O15350.1/117-309 | 20.84 | 23.11 | 54.67 | 42.55 | 35.76 | 42.44 | 47.59 | 40.35 | 52.76 | 49.45 | 45.41 | 41.44 | 57.19 | 84.73 | 100.00 |

**Fig 2. Sunburst plot of the distribution of the p53 protein domain (PF00870) in living organisms according to Pfam (accessed September 06, 2021).** The plot shows the distribution of the 1,765 sequences containing the P53 binding domain across 382 species. Every bar on the periphery represents one single species, containing one or several p53 paralogues in their genome (**A**), and percent identity matrix created by CLUSTALW 2.1 of the 7 human p53 domain-containing proteins and two of their most distant homologues from the choanoflagellate *Monosiga brevicollis* (**B**).

amino acids. According to Pfam 35.0 [27], HsTP53 contains four main protein domains: P53 TAD (transactivating domain), TAD2, P53 DNA binding domain, and P53 tetramer. P63 and p73 also contain the P53 DNA binding domain and the P53 tetramer domain. The P53 TAD and TAD2 domains are absent in P63 and P73, but instead both include a single SAM_2 domain.

P53 (PF00870 in Pfam) is the main domain of the p53 protein, covering the amino acids 99 to 289. Pfam contains 1765 P53-domain-containing sequences from 382 species, all in choano-organisms (metazoans and choanoflagellates), including 5 sequences in choanoflagellates and 13 sequences in the genome of *Homo sapiens* (**Fig 2A**, the figure can also be accessed here: https://pfam.xfam.org/family/PF00870#tabview=tab7). P53 TAD and TAD2 are two transcription scaffold domains. The Pfam database [27] includes 253 sequences containing the P53 TAD domain, in bilaterians only. The domain TAD2 is present in 81 sequences, from primates only. P53 tetramer serves for the oligomerization of the protein. The database includes 1,392 sequences, in animals only, containing the domain p53 tetramer. The SAM 2 (sterile alpha motif) domain is a putative protein interaction domain. More than 20,000 sequences containing this domain are present in Pfam [27], in more than 1,400 species.

The SCOP classification of p53 is as follows (accessed September 06, 2021):

- **Class b**: all beta-proteins. This class contains 178 folds.

- **Fold b.2**: common fold of b.2: Common fold of diphtheria toxin/transcription factors/cytochrome f. This fold contains 9 superfamilies.

- **Superfamily b.2.5**: p53-like transcription factors. This superfamily contains 8 families.

- **Family b.2.5.2**: p53 DNA-binding domain-like. Three proteins belonging to this family are present in the database.

- **Protein p53** tumor suppressor, DNA-binding domain. The p53 proteins of 2 species are present in SCOP: *Homo sapiens* and *Mus musculus.*

**2. Identification of homologues.** To reconstruct the evolutionary history of the p53 domain in animals, the selection of TP53 homologues covering the diversity of the family is necessary.

*Using a protein BLAST (Blastp) (https://blast.ncbi.nlm.nih.gov/Blast.cgi?PAGE=Proteins), paste the sequence of HsTP53 in the FASTA format, and select the genomes of the species of interest. Launch a BLAST search and download the amino-acid sequences in the FASTA format. Then, paste all the sequences in a single file using '.fasta' as filename extension.*

In this example, the p53 homologues of diverse animals (the cnidarian *Hydra vulgaris*, four insect species: *Drosophila melanogaster*, *Apis mellifera*, *Bombus terrestris* and *Aedes aegyptus*, and the tunicate *Ciona intestinalis*) and the p53, p63, and p73 of diverse vertebrates (the teleost fish *Danio rerio*, the coelacanth *Latimeria chalumnae*, the amphibian *Xenopus tropicalis*, the lizard *Anolis carolinensis*, the bird *Gallus gallus*, and the mammals *Bos taurus* and *H. sapiens*) were chosen. The p53 of the choanoflagellate *Monosiga brevicollis* (a protist related to animals), also retrieved from a BLAST search was chosen as outgroup (S1 **File**).

**3. Multiple sequence alignment and alignment trimming.** *Use an alignment tool (e.g. MAFFT, https://www.ebi.ac.uk/Tools/msa/mafft/ [45]) to align the sequences. Paste the alignment in the FASTA format and submit. Save the alignment in a new FASTA file. You can also directly download the sequences into the Guidance 2 server (http://guidance.tau.ac.il/) [68] and proceed to the alignment using MAFFT. Open the color-coded MSA to identify poorly aligned and highly variable regions. You can delete them manually from the alignment or remove unreliable columns below a certain cutoff. The new MSA, hereafter renamed sub-MSA, will be used for the phylogenetic analysis.*

*Optional: calculate the identity matrix of the sequences using alignment tools (e.g., CLUSTALW 2.1).*

The 13 human p53 paralogues share 36% to 100% identity, and the two paralogues of *Monosiga brevicollis* share 21.4% identity (**Fig 2B**). Human and *Monosiga* orthologues share 17% to 25% identity. Hence, all human paralogues are more similar to each other than to any of the *Monosiga* orthologues.

**4. Sequence evolution model selection.** We propose to perform a phylogenetic analysis of the p53 family using a distance-based method (NJ) and a probabilistic method (ML) and compare the results. First, it is necessary to identify the optimal model of sequence evolution.

*Here, we are using protein sequences. Use ProtTest 3.4.2 [100] to calculate the log-likelihoods of a panel of 56 amino-acid substitution models, and select the most relevant one based on the BIC or AIC score. Select the model with the lowest score.*

*Alternatively, use the substitution model selectors included in IQ-TREE or MEGA. For example, with the IQ-TREE web server (http://iqtree.cibiv.univie.ac.at/), open the Model Selection panel, download the sub-MSA, select "protein sequences", choose a selection criterion (AIC or BIC) and proceed to the analysis. With MEGA 11, download the sub-MSA, and select "Find best DNA/protein models" in the Model panel.*

The model JTT+G [90] (JTT with Gamma-distributed rate-heterogeneity across sites), that minimizes the BIC score, was selected.

**5. Phylogenetic inference.** For beginners, we recommend using programs that include a user interface or an online version, such as MEGA, SeaView, or the IQ-TREE server. The phylogenetic trees were inferred using MEGA 11 [55] for the NJ-based analysis, and IQ-TREE 2 [77, 78] for the ML-based analysis, using the sub-MSA and the appropriate model (JTT+G).

*With MEGA 11, in the Phylogeny panel, perform a phylogenetic analysis using NJ with the sub-MSA. Select the appropriate substitution model (e.g., JTT+G) and the bootstrap method with e.g., 1000 replicates.*

*With IQ-TREE 2, download the alignment file, select the appropriate sequence type (DNA or protein) and the appropriate substitution model (e.g., JTT+G). In the panel "branch support analysis", select the Ultrafast Bootstrap analysis with e.g., 1000 replicates. For single branch tests, you can also select the SH-aLRT test.*

*Save the phylogenetic tree including the Bootstrap/SH-aLRT values and branch lengths in the Newick format and open it with FigTree or ITOL for a graphical display of the tree. You can also paste the tree in the Newick format directly into the graphical program.*

Both methods reveal four major clades containing respectively the p53 of insects and the p53, p63, and p73 of all vertebrates (**Fig 3**). The p53, p63, and p73 of vertebrates are more closely related to each other than to any other p53. Furthermore, the p63 and p73 of vertebrates are more closely related to each other than to vertebrate p53. This indicates that two duplication events in the p53 family preceded the origin of vertebrates. First, the p53 family and the p63/p73 cluster diverged. The second one caused the p63 and p73 families to diverge (**Fig 3**). The p53 of insects are clustered together. This indicates that insects diverged from the other bilaterians before these two duplications. These results are in accordance with the existing literature on the evolutionary history of the p53 family [190].

**6. Molecular dating of speciation events.** We propose to estimate the age of speciation and duplication events of our phylogenetic tree. TimeTree has been used to retrieve the estimates of age of speciation events, and these events were used as calibration points. Molecular clocks can also be used to calibrate the phylogeny in MEGA.

*Download the alignment file in the FASTA format and the phylogenetic tree in the Newick format in MEGA 11. In the* Compute *panel, select "Compute TimeTree" and "internal nodes constraints". In TimeTree (http://www.timetree.org/), enter the names of two species of interest. For example,* Homo *and* Drosophila *diverged between 630 and 830 million years ago, with 694 million years as median time. In MEGA, click "add new calibration point" and select the node in the phylogenetic tree, or enter the names of the two taxa, and define the speciation age with a minimum, maximum or fixed time (for example, 694 million years between* Homo *and* Drosophila*). Use TimeTree to define several calibration points, before and after the duplication events, and save the calibrated tree.*

A

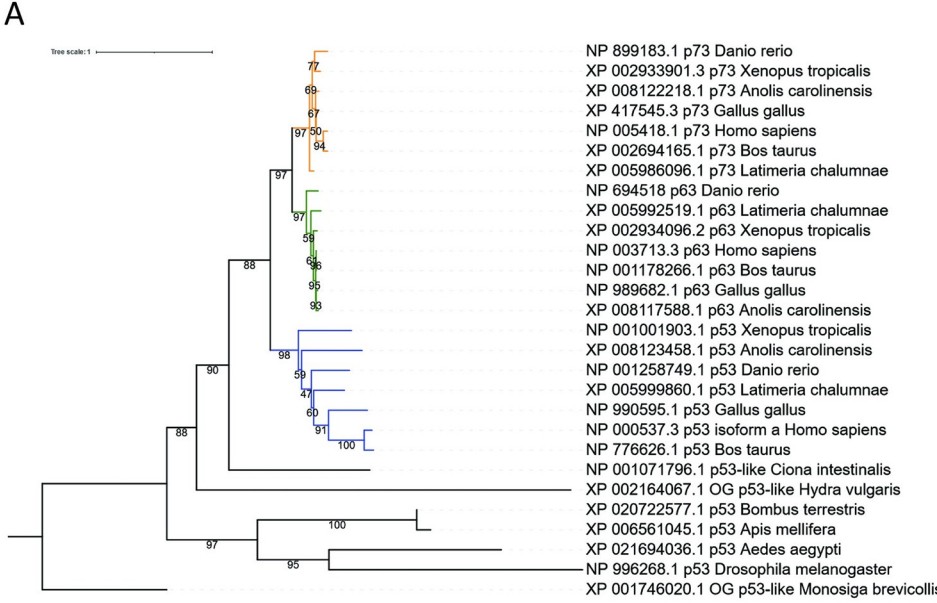

B

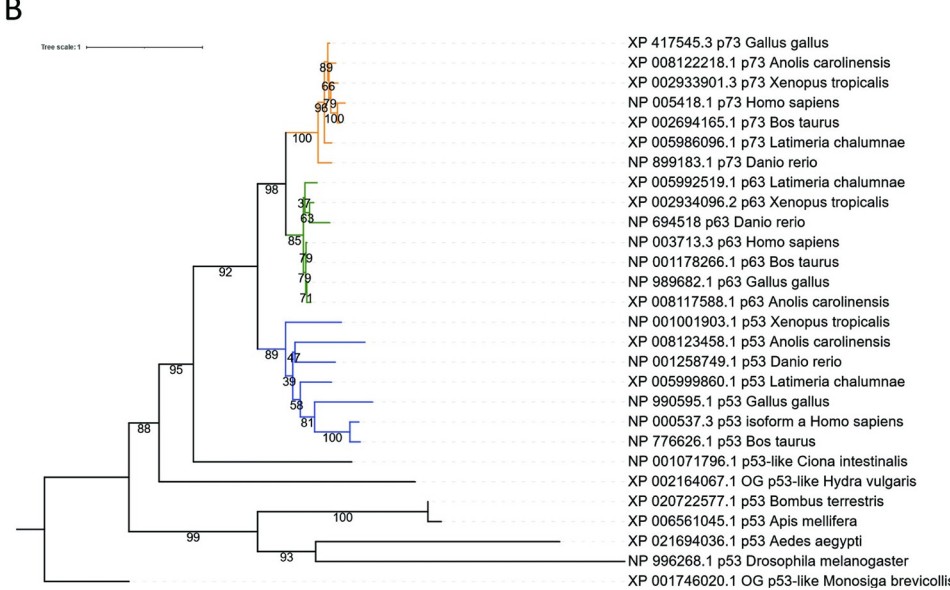

**Fig 3.** Phylogenetic trees of the P53 domain-containing proteins of metazoans using Neighbor Joining (A) and Maximum Likelihood (B). The trees were realized according to the model JTT+G [90], as calculated by ModelFinder [101], using the AIC [98]. The numbers on the internal edges/branches indicate the bootstrap values as calculated by the standard bootstrapping method [138] and UFBoot2 [131], respectively. The phylogenetic trees were inferred using MEGA 11 [55] and IQ-TREE 2 [77, 78], respectively, and the figures were generated using ITOL [153]. Green branches represent the p63 family, orange branches represent the p73 family and blue branches represent the p53 family.

The time-calibrated phylogeny of the TP53 family suggests that the duplication event between p53 and the p67/p73 cluster occurred around 502 million years ago, and that p63 and p73 diverged 452 million years ago (**Fig 4**). One should keep in mind that these evolutionary ages are only estimates based on a few calibration points. According to the database Ohnologs [171], p53, p63 and p73 result from the two-round whole genome duplication event that preceded the origin of chordates.

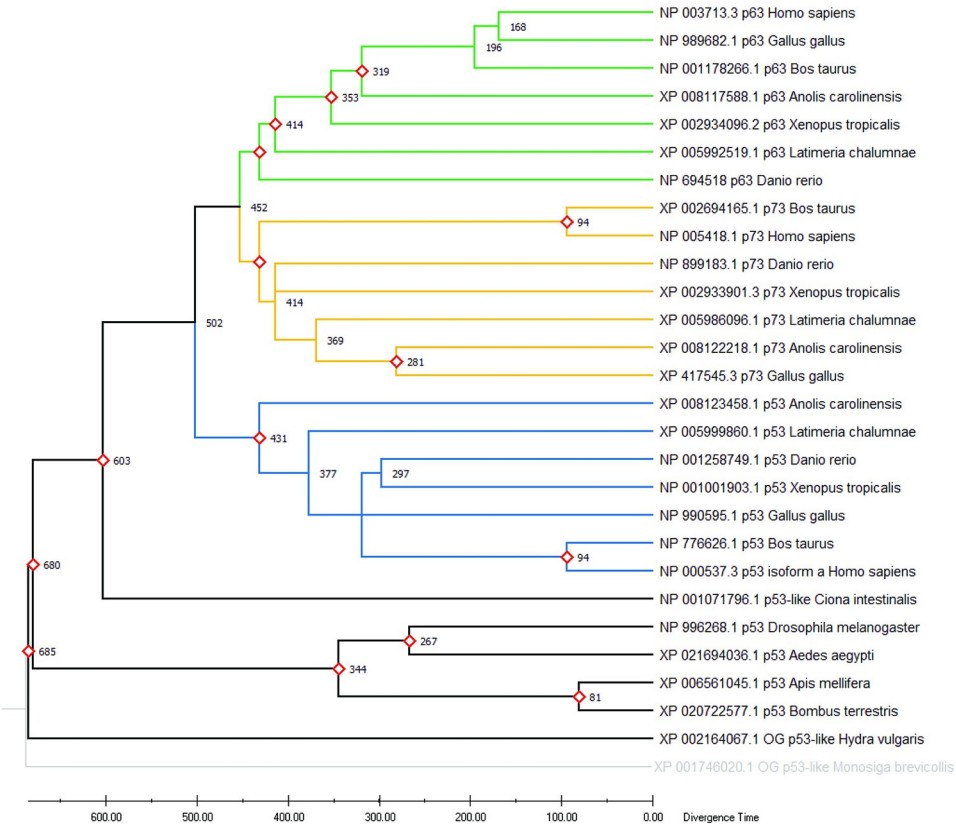

**Fig 4. Time-calibrated phylogenetic tree of the P53 domain-containing proteins of metazoans.** The trees were realized according to the model JTT+G [90], as calculated by ModelFinder [101] using the AIC [98]. The phylogenetic tree and the figure were realized using MEGA 11 [55]. Time calibration was performed using TimeTree [173]. The values at the nodes and the scale indicate the divergence time in million years. Green branches represent the p63 family, orange branches represent the p73 family and blue branches represent the p53 family. Gray spots on the branches indicate the appearance of the different protein domains during the evolution of the TP53 family.

**7. Reconstruction of the evolutionary history of the P53 family.** By combining this phylogenetic tree and the database Pfam [27], the evolutionary history of the protein family can be traced. The p53 DNA binding domain, shared by all proteins in this analysis, appeared before the divergence between choanoflagellates and metazoans (**Fig 4**). The SAM2 domain, present in p63 and p73 sequences of vertebrates only, appeared after the p53-p63/p73 duplication and before the p63-p73 duplication. The P53 TAD domain is restricted to vertebrate p53. It appeared after the first whole genome duplication and before the speciation of vertebrates. Finally, the TAD2 domain evolved recently and is restricted to primate p53.

## Evolutionary history of human cyclins and CDKs

Cyclin-dependent kinases (CDKs) are protein kinases involved in the control of cell cycle. They are responsible for the activation of specific target proteins. CDKs are activated by regulatory proteins called cyclins, that are characterized by a cyclic variation of the concentration along the cell cycle. Cyclin binding to the CDK activates specific kinases and phosphatases that in turn activate the CDK. Subsequent ubiquitination and proteolysis of cyclins by the anaphase promoting complex then inactivate the CDK. All steps of the cell cycle (mitosis, G1, G2, S) depend on the activation of specific CDKs by specific cyclins. Cyclins and CDKs represent large families of proteins. Twenty-one CDKs and twenty-one cyclins are present in the human

genome. Here, we present a protocol for studying the evolutionary history of human cyclin and CDK paralogues and their coevolution, following the roadmap presented above.

**1. Phylogenetic analyses.** To study the coevolution between human cyclins and CDKs, we need first to reconstruct their phylogenies separately. In this example, two human proteins related to CDKs, GSK3 and MAK, were chosen as outgroups for CDKs [191] (**S2 File**). Cables1 and Cables2, related to cyclins, were used as outgroups for the phylogenetic analysis of cyclins [191] (**S3 File**). Sequences were aligned using CLUSTALW 2.1 [39] and ModelFinder [101] has been used to determine the most relevant evolutionary model based on the AIC. The amino acid substitution model LG [192] has been selected for both families. Then, phylogenetic analysis was performed using maximum likelihood with IQ-TREE 2 [77, 78] and consistency of the phylogenetic estimate was assessed using the bootstrapping method UFBoot2 [131]. The figures were generated using ITOL [153].

**2. Identification of homologues resulting from whole genome duplications.** Using the phylogeny (**Fig 5**) and the database Ohnologs 2.0 [171], ohnologues, *i.e.*, paralogues resulting from a whole genome duplication, can be identified (**Fig 5**). For example, in the cyclin family, Cyclins T1 and T2, Cyclins B1 and B2, Cyclins A1 and A2, and Cyclins E1 and E2 are ohnologues. In the CDK family, CDK12 and CDK13, CDK4 and CDK6, CDK14 and CDK15, and CDK19 and CDK8 are ohnologues. These ohnologues likely resulted from two-round whole genome duplication that occurred before the origin of chordates [193, 194].

**3. Study of the coevolution between cyclins and CDKs.** To study the coevolution between the two gene families, the phylogenetic trees of cyclins and CDKs and their associations are needed. **Jane 4** [170] and **Treemap 3** [174], two programs designed for studying coevolution between hosts and parasites, were used to reconstruct the co-phylogeny of the two gene families. This example uses the cyclin/CDK associations from a publication on the evolution of the Cyclin and CDK families [195].

*With Jane and TreeMap, a single nexus file containing the phylogenies of cyclins and CDKs, and their associations is needed. Create a nexus file (starting with #NEXUS). This file should contain the two trees in the Newick format, in the sections BEGIN HOST and BEGIN PARASITE, and the associations in the section BEGIN DISTRIBUTION. This section should mention every association between Cyclins and CDK following the pattern "Host: Parasite,". All three sections should end with "ENDBLOCK;". The names of the taxa in the three files should be identical. Cyclins interacting with several CDKs and vice-versa should be repeated (S4 File).*

*Import this file to Jane and launch the analysis in the Solve Mode. The costs of coevolutionary events can be set. The stats mode can be used to compute the cost range of the solutions. With TreeMap, import the nexus file and launch the analysis in "Solve the tanglegram". We obtain a coevolutionary scenario that represents the best way to associate the two trees. You can test the significance of the reconstruction in "estimate significance" or perform a heuristic test.*

The co-phylogenies of cyclins and CDKs are presented in **Fig 6**. These figures retrace the evolutionary history of cyclins and CDKs. Several coevolution events were identified, including co-speciation, duplication, duplication with interaction switch, loss of interaction, and numerous failures to diverge, for example the duplication of Cyclins L1 and L2 without a duplication in CDK9 (**Fig 6A**). Significant co-evolution events are identified by both programs, such as between the 3 Cyclin D paralogues and the CDK4 and CDK6 cluster, and between Cyclins A, B, D and E and CDK 1, 2, 3, 4, 6, 14 and 16 (**Fig 6A and 6B**).

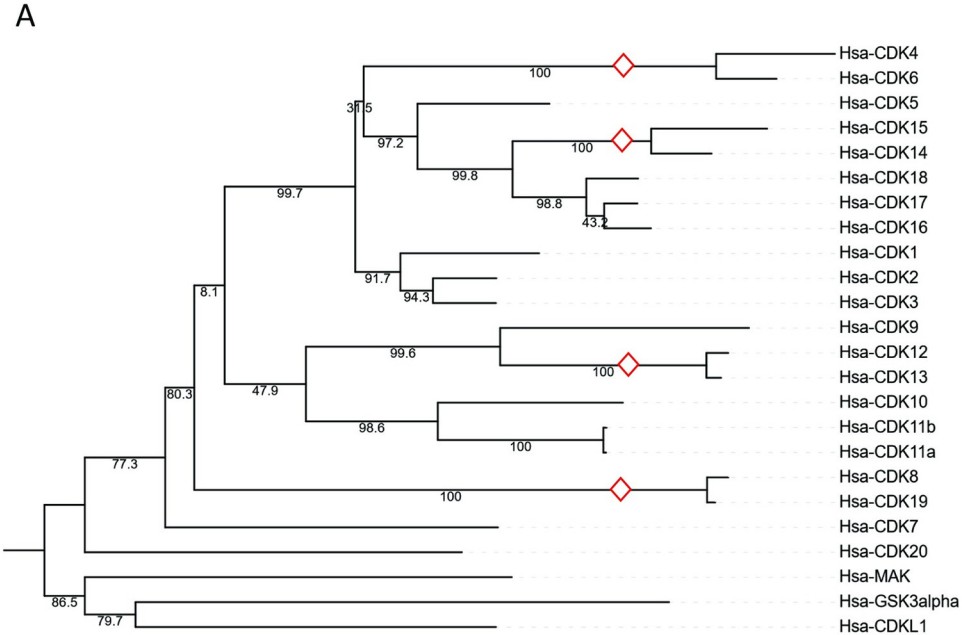

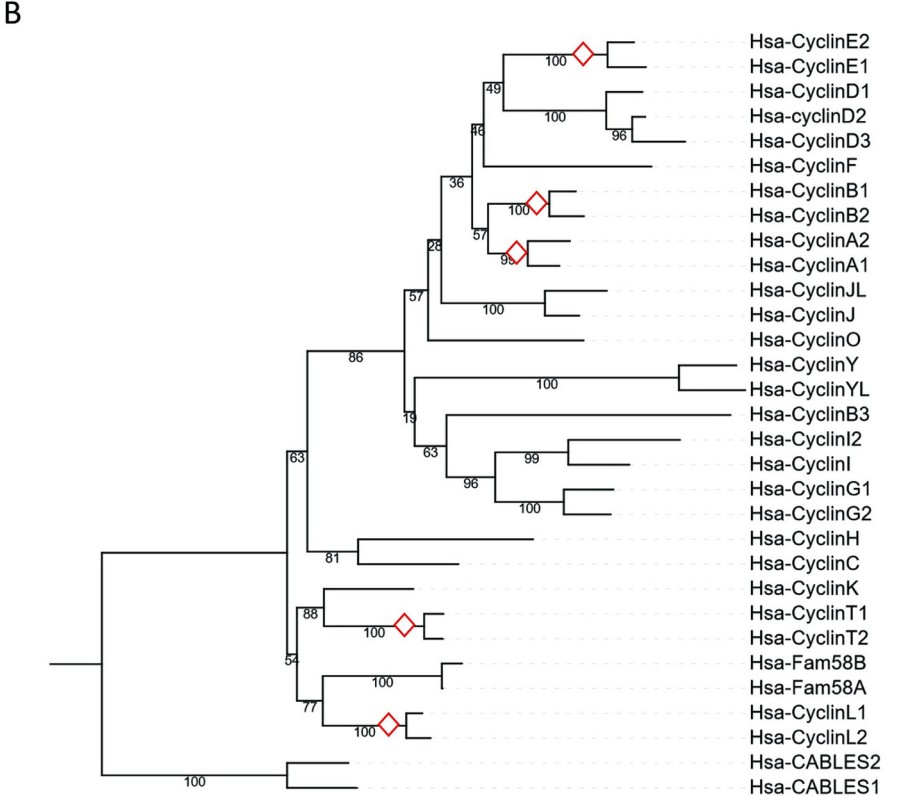

**Fig 5.** Phylogenetic trees of human CDKs (A) and Cyclins (B) using Maximum likelihood. The trees were realized according to the substitution model LG [192] as calculated by ModelFinder [101] using the AIC [98]. The phylogenetic analyses were performed using IQ-TREE2 [77, 78] and the figure was realized using ITOL [153]. The numbers indicate the bootstrap values as calculated by UFBoot2 [130]. Red squares indicate whole genome duplications according to the database Ohnologs 2.0 [171].

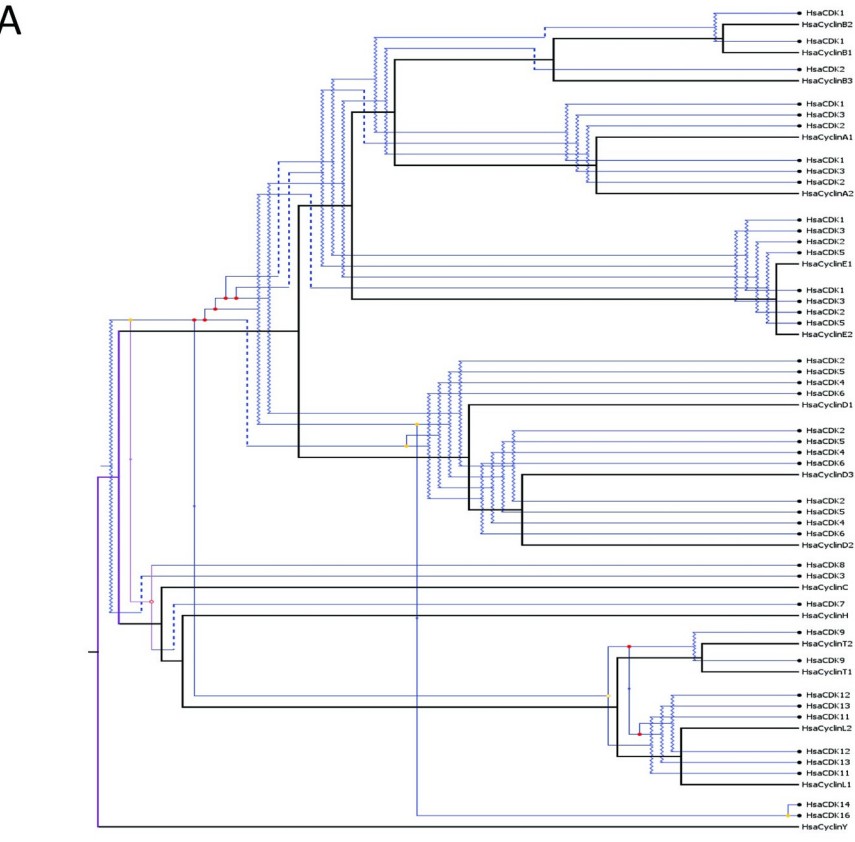

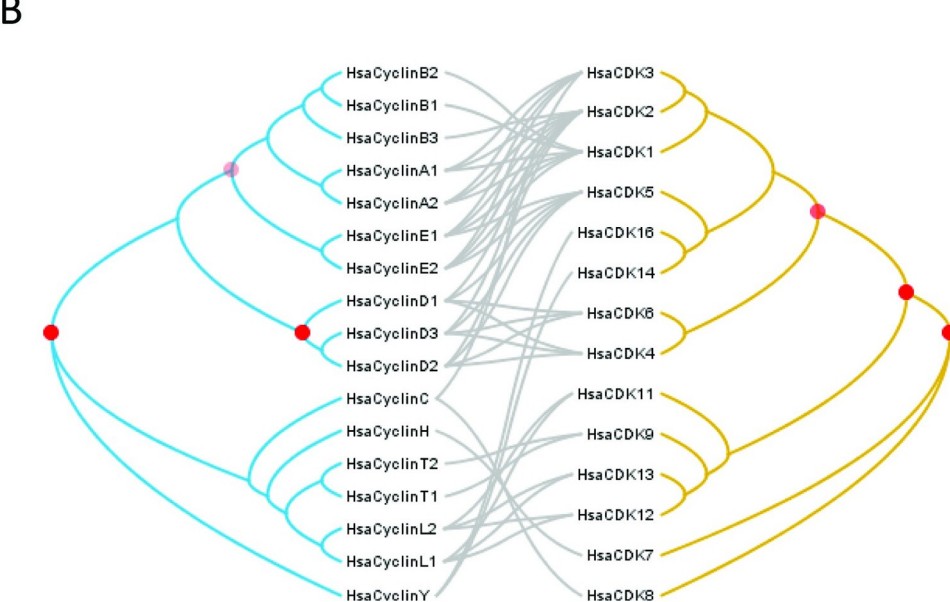

**Fig 6. Two co-evolutionary scenarios between human cyclins and human CDKs.** The co-phylogenies were realized using Jane [170] (**A**) and TreeMap [174] (**B**). (**A**) Cyclins (black lines) and CDKs (blue lines) that cluster together depict an interaction (the cyclin can bind the CDK to activate it). Hollow red circles indicate co-speciation events, solid red circles indicate a duplication, and yellow circles indicate a duplication with host switch. Dashed lines illustrate a loss of interaction, and jagged lines indicate a failure to diverge. (**B**) Significant co-speciation events between cyclins and CDKs are indicated by red filled circles (graded). The more intense red color indicates a more significant congruence.

## Supporting information

**S1 File. Sequences and accession numbers of p53 proteins.** These sequences and accession numbers were used for phylogenetic analysis.
(PDF)

**S2 File. Sequences and accession numbers of human CDKs.** These sequences and accession numbers were used for phylogenetic analysis.
(PDF)

**S3 File. Sequences and accession numbers of human cyclins.** These sequences and accession numbers were used for phylogenetic analysis.
(PDF)

**S4 File. Nexus code.** The coding was used to reconstruct the co-phylogeny of human CDKs and Cyclins.
(PDF)

**S5 File. Protocol.** The protocol as also available on protocols.io.
(PDF)

## Acknowledgments

We are grateful to Sarah Amend, Kenneth Pienta, and Laurie Kostecka at the Brady Urological Institute, Johns Hopkins School of Medicine, and to Stina Andersson, Chris Carroll, and Sinan Karakaya at the Tissue Development and Evolution (TiDE) group, Lund University, for carefully reading the manuscript and providing useful comments that improved the paper.

## Author Contributions

**Conceptualization:** Florian Jacques, Emma U. Hammarlund.

**Data curation:** Florian Jacques.

**Funding acquisition:** Emma U. Hammarlund.

**Investigation:** Florian Jacques, Paulina Bolivar.

**Methodology:** Florian Jacques, Paulina Bolivar.

**Project administration:** Kristian Pietras, Emma U. Hammarlund.

**Software:** Florian Jacques, Paulina Bolivar.

**Supervision:** Kristian Pietras, Emma U. Hammarlund.

**Writing – original draft:** Florian Jacques, Emma U. Hammarlund.

**Writing – review & editing:** Florian Jacques, Paulina Bolivar, Kristian Pietras, Emma U. Hammarlund.

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
