## [Decision Letter · Decision Letter 0]

23 May 2022

PONE-D-22-06715Roadmap to the study of gene and protein evolutionPLOS ONE

Dear Dr. Jacques,

Thank you for submitting your manuscript to PLOS ONE. After careful consideration, we feel that it has merit but does not fully meet PLOS ONE’s publication criteria as it currently stands. Therefore, we invite you to submit a revised version of the manuscript that addresses the points raised during the review process.

We look forward to receiving your revised manuscript.

Kind regards,

Arndt von Haeseler

Academic Editor

PLOS ONE

Journal Requirements:

4. PLOS requires an ORCID iD for the corresponding author in Editorial Manager on papers submitted after December 6th, 2016. Please ensure that you have an ORCID iD and that it is validated in Editorial Manager. To do this, go to ‘Update my Information’ (in the upper left-hand corner of the main menu), and click on the Fetch/Validate link next to the ORCID field. This will take you to the ORCID site and allow you to create a new iD or authenticate a pre-existing iD in Editorial Manager. Please see the following video for instructions on linking an ORCID iD to your Editorial Manager account: https://www.youtube.com/watch?v=_xcclfuvtxQ.

Additional Editor Comments:

The ms has been evaluated by two experts in the field. I do agree with the reviewers remarks and would like to ask you to address the critcism and to rewrite

the ms accordingly.

Reviewers' comments:

Reviewer's Responses to Questions

**Comments to the Author**

1. Does the manuscript report a protocol which is of utility to the research community and adds value to the published literature?

Reviewer #1: Yes

Reviewer #2: Yes

2. Has the protocol been described in sufficient detail?

Descriptions of methods and reagents contained in the step-by-step protocol should be reported in sufficient detail for another researcher to reproduce all experiments and analyses. The protocol should describe the appropriate controls, sample sizes and replication needed to ensure that the data are robust and reproducible.

Reviewer #1: Partly

Reviewer #2: No

3. Does the protocol describe a validated method?

Reviewer #1: Yes

Reviewer #2: No

4. If the manuscript contains new data, have the authors made this data fully available?

Reviewer #1: N/A

Reviewer #2: N/A

**5. Is the article presented in an intelligible fashion and written in standard English?**

Reviewer #1: Yes

Reviewer #2: **No: **The manuscript can be written more succinctly

6. Review Comments to the Author

Reviewer #1: In this manuscript, Jacques et al. present a guideline for studying gene and protein evolution. In summary, several available public resources for DNAs and proteins are introduced, which provide the sequence data and their related characters such as gene functions, protein structures, the associated biological pathways or diseases, etc. They can be specific, focusing only on a particular model organism such as TAIR, WormBase or The human protein atlas, or can also be very widespread like NCBI or UniProt. Next to those databases, the authors listed different tools and methods for reconstructing and analyzing the phylogenetics of genes or proteins, starting from the alignment of the homologous genes to tree building and visualizing the result trees. They furthermore highlighted some applications that can be utilized from the phylogenetic analysis, such as measure of selection strength, study of co-evolution, or genome evolution, etc. Finally, the manuscript is completed with two tutorials to reconstruct the evolutionary history of p53 protein family and the coevolution between human cyclins and CDKs.

Overall, this manuscript provides a practical protocol for biologists to design a basic analysis of gene evolution. Still, I see some issues and improvements.

Major issues:

1. In the sequence alignment session, the authors already mentioned that ClustalW is suitable for large dataset, while MAFFT has the highest accuracy. Why hasn’t MAFFT been used in the two examples instead of Clustal2.1, even though the dataset was small enough for MAFFT? How will it affect the result by using different alignment tool?

2. Both two examples considered small datasets. For example, the reconstruction of the evolutionary history of p53 family has been done with only 11 taxa. It is, therefore, doable by either manually typing the species to the search organism field or downloading protein sequences for each species from the online BLAST search result. However, in many cases, especially while studying about the evolutionary history of a gene, one needs a large and more diverse taxa set (i.e. https://doi.org/10.3389/fmicb.2021.739000). It would be appropriate to have an instruction for effectively batch downloading the required data. Or, in general, how to deal with retrieving large data?

3. The databases are intensively represented. The information, such as the gene or protein sequences, can be redundant between databases. Nevertheless, their identifiers are often different. It would be helpful to quickly outline the approaches to communicate between different resources, for example by using the ID mapping tool of UniProt (https://www.uniprot.org/uploadlists/).

4. Similar to the issue with the databases, a large number of tools has been introduced. However, many of them are just superficially described, especially the tools for the phylogenetic analysis in table 6 and 7. Thus, this can puzzle the users for choosing the right approach for their study, which consequently defeats the purpose of this manuscript.

Minor issues:

1. FigTree should not belong to the list of nucleic acid databases (table 1).

2. Link for UniProt in table 2 should not be directed to a specific protein (P04637) but be generalized.

3. Gblocks Server is no longer available (link in table 4).

4. Links for BayesTraits, HGT-Finder in table 6 and Bio++ in table 7 do not work (tested on May 15th).

5. Link for PyMol in table 8 is directed to Mesquite.

6. The reference for some tools are missing in the main text and tables. Namely, ClustalOmega, Probcons (table 4), BayesTraits, FigTree, PAUL (table 6), and Bio++ Suite (table 7).

Reviewer #2: GENERAL COMMENTS

In general, I think there is merit in publishing a manuscript like the one submitted. However, I also think you could make it more comprehensive and more accurate, especially regarding the section on the protocol and phylogenetic analysis.

Regarding the protocol, I think readers would need to have a more detail decision tree that offers them alternative paths, depending on what their objectives and data are. You need to incorporate some measures of quality control at different steps in the protocol, and you need feedback loops that are followed in case the result of a quality control is that the preceding analysis did not yield the expected result (otherwise you will perpetuate errors made at the early steps). In this regard, your protocol is a little like that in Ciccarelli et al. (2006; Science 311, 1283-1287).

I welcome that you cite a lot of databases, but could you move their citations from where they are to right after the name? Sometimes, a citation is at the end of the sentence and might be mistaken for a citation to something else (e.g., other databases or multiple sequence alignment).

Regarding the phylogenetic analysis (L194-L364), I compliment you on a valiant attempt to cover a large and complex field. However, I do not think you succeeded because I found gaping holes:

1. You seem unaware of previous phylogenetic protocols, one of which appeared recently in NAR Genomics & Bioinformatics (2, lqaa041).

2. Because the manuscript does not present something novel but summarises bioinformatics tools and resources, chiefly databases, you need to be comprehensive. Unfortunately, you were not comprehensive. This applies to both multiple sequence alignment methods and phylogenetic methods.

3. Currently, I am comparing the accuracy of 20 multiple sequence alignment methods (i.e., Clustal Omega, CONTRAlign, DIALIGN-TX, Fsa, GramAlign, KAlign, MAFTT (default), MAFFT (EINSI), MAFFT (GINSI), MAFFT (LINSI), MSAProbs, Muscle, Pasta, PicXAA, Poa, Prank, ProbAlign, ProbCons, T-Coffee (fast), and T-Coffee (regressive)). You considered only a few of these.

4. You mention one method for trimming sites from multiple sequence alignment (GBlocks; in Table 4). There is a suite of other and more suitable methods available (see NAR Genomics & Bioinformatics 2, lqaa024; see also citations 13-21 in that paper).

5. You mention three model-selection methods but overlooked other more flexible methods (Nature Methods 14, 587-589; Mol. Biol. Evol. 29, 1695-1701; Syst. Biol. 63, 726-742; Mol. Biol. Evol. 34, 772-773; Syst. Biol. 69, 249-264).

6. Your understanding of the relationship between log-likelihood and the AIC and BIC is wrong (L232-L233), suggesting confusion. You should read Briefings in Bioinformatics (21, 533-565).

7. Your understanding of what the non-parametric bootstrap scores indicate is wrong (see Science 310, 1911-1912).

8. You mention several distance matrix-based phylogenetic methods but do not cite papers describing them.

9. Your list of phylogenetic methods is incomplete and misses many important programs (e.g., IQTREE (Mol. Biol. Evol. 32, 268-274), IQTREE2 (Mol. Biol. Evol. 37, 1530-1534), PHYLIP (Felsenstein, 2005), PyCogent (Genome Biology 8, R171), FastME v2.0 (Mol. Biol. Evol. 32, 2798-2800), LSD (Syst. Biol. 65, 82-97) Garli (Syst. Biol. 63, 812-818), PhyloBayes 3 (Bioinformatics 25, 2286-2288), SplitsTree (Mol. Biol. Evol. 23, 254-267)).

10. You really need to ensure that software and methods referred to are cited properly, preferentially every time and with version numbers included.

11. Your figures are unclear, and the colours used are not consistent with a colour palette fit for colourblind people.

SPECIFIC COMMENTS

L47-L48. Why start with “Although … community”?

L61. “allowed” > “has allowed”

L61-L62. “nucleic acid and protein” > “nucleotides and amino acids” [DNA is a sequence of nucleotides and protein is a sequences of amino acids; therefore, it is not correct to write DNA sequences or protein sequences]

L62. “In addition, this data is” > “Typically, these data are” [datum is singular of data]

L66-L67. “navigate these resources can be puzzling” > “navigate through these resources can be challenging”

L81. Delete “considerable accompanying”

L84-L86. The figure is not a fully developed protocol like those in NAR Genomics & Bioinformatics 2, lqaa041 [2020].

L89. “DNA sequences” > “nucleotide sequences (DNA)” [I am pedantic here, but DNA is a sequence (of nucleotides) so writing DNA sequences is the same as saying a sequence of a sequence of nucleotides]

L89. “and their protein translations” > “and, where applicable, their translations into protein”

L103. FigTree is not a database!!! It is a phylogenetic tool, so it should be in a different table.

L119. “protein sequences” > “amino-acid sequences” [I use a hyphen because amino and acid form a compound adjective]

L123. See above.

L148. “sequences alignment” > “pairwise sequence alignment” OR “multiple sequence alignment”

L159. “evolutionary relationships” > “evolutionary relationships (see below)”

L167 “under FASTA” > “in FASTA”

L177. “, i.e., … ancestry” > “(i.e., … ancestry)” [I prefer “that is,” in the sentence and “i.e.,” inside parentheses]

L177-L179. This sentence should be so that the terms homology, orthology, paralogy and xenology are clear and unambiguous.

L184. Include “(define/explain the E-value)”

L185. “amino acid sequences” > “amino-acid sequences”

L186. Explain “HH”

L196. “proteins within” > “proteins, even within”

L197. “data, (“ > “data (“

L198. “), includes” > “) includes”

L202. Delete “sequences”

L203. “one after the other” > “, one after the other,”

L213-L214. Not “distantly related”; “closely related” [I spoke to Ary and Nick about this]

L227. You list the two least realistic models of rate heterogeneity across sites. More realistic models of rate heterogeneity across sites are described in Kalyaanamoorthy et al. (Nature Methods 14, 587-589) and Crotty et al. (Syst. Biol. 69, 249-264).

L232-L233. This sentence is incorrect. The AIC and BIC are computed from the log-likelihood scores; not the other way around, as you write.

L235-L239. Your description of the reason for doing a non-parametric bootstrap analysis is misleading and is not supported by relevant literature (e.g., Mol. Biol. Evol. 9, 366-369; Mol. Biol. Evol. 30, 1188-1195; Mol. Biol. Evol. 35, 518-522).

L247-L250. Include citations after each method or program mentioned, please.

I will stop my general review here, but urge you to revise every page in accordance with the suggestions given above.

7. PLOS authors have the option to publish the peer review history of their article (what does this mean?). If published, this will include your full peer review and any attached files.

Reviewer #1: No

Reviewer #2: No

---

## [Author Response · Author response to Decision Letter 0]

24 Aug 2022

We thank the reviewers for their helpful and insightful comments. We have revised the manuscript in accordance with most of their comments and concerns. We believe that their thorough review and our revision has resulted in a much-improved contribution. Below, we reply to all the comments one by one. We refer to new intervals of text by their row number in the manuscript.

Reviewer 1:

1. In the sequence alignment session, the authors already mentioned that ClustalW is suitable for large dataset, while MAFFT has the highest accuracy. Why hasn’t MAFFT been used in the two examples instead of Clustal2.1, even though the dataset was small enough for MAFFT? How will it affect the result by using different alignment tool?

Yes, we agree and thank the reviewer for pointing it out. We used MAFFT in our first example and ClustalW in the second example, because they are two of the most user-friendly for beginners since they have a web interface. In our first example, ClustalW has been used only to generate the percent identity matrix. In the introduction we have now also clarified the overall aim to highlight the databases and tools that display a web-user interface. In our examples, the differences in results (topology tree and nodes robustness) with MAFFT or ClustalW is minor. Although a comparison of the different results could be interesting, we refrain from discussing it here since it is beyond the scope of this guide to the practical tools. More specifically, edits are made at

Row 69-70: ‘The aim is to provide a practical guide for beginners and more advanced explorers into protein and gene evolution.’

Row 233-235: ‘ PROBCONS [42], T-COFFEE [41] and MAFFT [40] are described to have particularly high accuracy but also high calculation times [45]. They are suitable for small and intermediate datasets.’

2. Both two examples considered small datasets. For example, the reconstruction of the evolutionary history of p53 family has been done with only 11 taxa. It is, therefore, doable by either manually typing the species to the search organism field or downloading protein sequences for each species from the online BLAST search result. However, in many cases, especially while studying about the evolutionary history of a gene, one needs a large and more diverse taxa set (i.e. https://doi.org/10.3389/fmicb.2021.739000). It would be appropriate to have an instruction for effectively batch downloading the required data. Or, in general, how to deal with retrieving large data?

Yes, this is a valid comment. We have now revised the text to clarify how to extract a large number of sequences from NCBI using different identifiers, and how to save them into a fasta file. Also, we now introduce the Batch Entrez tool. More specifically, edits are made at

Row 121-123: ‘It is also possible to batch download a large number of sequences from NCBI, by entering their identifiers (accession numbers, gi numbers or GeneIDs) in Batch Entrez.’

Row 133-134: ‘The “retrieve/ID mapping” tool of UniProt allows to batch download the information on a list of proteins using UniProt identifiers.’

3. The databases are intensively represented. The information, such as the gene or protein sequences, can be redundant between databases. Nevertheless, their identifiers are often different. It would be helpful to quickly outline the approaches to communicate between different resources, for example by using the ID mapping tool of UniProt (https://www.uniprot.org/uploadlists/).

Yes, we thank the reviewer also for pointing this out. We have revised the text to clarify how to communicate between the identifiers of several databases and introduced the ID mapping tool of Uniprot. More specifically, edits are made at

Row 130-132: ‘The “retrieve/ID mapping” tool of UniProt allows to batch download the information on a list of proteins using UniProt identifiers.’

4. Similar to the issue with the databases, a large number of tools has been introduced. However, many of them are just superficially described, especially the tools for the phylogenetic analysis in table 6 and 7. Thus, this can puzzle the users for choosing the right approach for their study, which consequently defeats the purpose of this manuscript.

We understand the reviewers concern here. There is a delicate balance when condensing a complex field into a practical guide that is aimed also for beginners (as well as somewhat more advanced users). We have tried to go into some more depth while clarifying how to choose the paths forward. More specifically, we have added several tools and bioinformatic methods for sequence alignment and trimming, model selection and phylogenetic inference, as well as comments on their specificities, including models of molecular evolution and bootstrapping methods. We edited the protocol, so it is more comprehensive, more clear and more user-friendly for non-bioinformatic users (and also stated this in the introduction). We detailed more the specificities of databases and tools in Tables 1, 2, 3, 4, 5, 6, 7, 8, 9, 10, and their corresponding paragraphs to facilitate the choice of appropriate method for the protocol users.

We explained more in detail the different methods for sequence alignments, and added several programs that can be used, with their specificities and comments on how to choose between them, mostly based on the size of the dataset. Specifically, edits are made on:

Row 232-243: ‘CLUSTALW [34] and MUSCLE [36] are included in MEGA [44]. They display web interfaces, as well as MAFFT [40], Kalign [39], and PRANKS [37,55]. PROBCONS [42], T-COFFEE [41] and MAFFT [40] are described to have particularly high accuracy but also high calculation times [45]. They should be restricted to small and intermediate datasets. CLUSTAL Omega [35] and Kalign [39] are particularly fast, but less accurate [46]. They can be used for datasets of up to 4000 and 2000 sequences, respectively [45,46]. The performances of MUSCLE are intermediate [46]. PRANK is particularly accurate for large sets and closely related sequences. Bali-Phy [47] performs a bayesian co-estimation of alignment, phylogeny, and other parameters and is also argued to be very reliable. PASTA [48] and UPP [49], that uses a machine learning technique, are designed for very large datasets. MAFFT offers a wide range of methods, which can be accuracy-oriented, such as L-INS-i, G-INS-i and E-INS-i; or speed-oriented, such as FFT-NS-2, which can be used for up to 30 000 sequences.’ 

We also added information on the different phylogenetic methods. They are now in separate paragraphs. We added several programs that can be used for phylogenetic inference, with some of their specificities and comments on how to choose between them, mostly based on the size of the datasets, but also their diverse options (type of data, models implemented, branch support test). Specifically, edits are made on:

Row 332-341 : ‘MEGA [44] and SeaView [104] are known to be very user-friendly. They include sequence alignment tools and tree editors. PhyML [105] is accurate, easy of use and, like PAUP [103] and MEGA [44], includes all common models of molecular evolution. RAxML [106] and particularly FastTree [100] are fast and well suited for large datasets (up to 1 million sequences with FastTree). They use a specific model of rate heterogeneity, in addition to Gamma law and proportion of invariant sites. IQ-TREE [60,61], that includes ModelFinder [84] and a very fast bootstrapping method, is reported to be both fast and accurate [111]. PAUP is slower than other programs and uses nucleotide data only.’

Row 345-349: ‘The main software used for BI-based phylogenetics is MrBayes [112] that uses the Markov Chain Monte Carlo (MCMC) algorithm (Table 7). PhyloBayes [113,114] is a bayesian MCMC sampler for phylogenetic reconstruction with protein data using a specific probabilistic model, well adapted for large datasets and phylogenomics. Bali-Phy [47] can also be used for phylogenetic analysis using Bayesian inference.’

To make the protocol more comprehensible and easier of use for beginner, we also added information on bioinformatic tools and how to use them, in the text and in the tables. More specifically, we added precisions on the tools used by the workflows.

Row 399-406: ‘The Cyberinfrastructure for phylogenetic research (CIPRES science gateway) [129] is a public resource for phylogenetic analysis that includes most tools and software for sequence alignment, model selection, and phylogenetic inference, including BEAST, FastTree, GARLI, IQ-TREE, jModelTest, MAFFT, MrBayes, PAUP, PhyloBayes and RAxML. Other packages include NGPhylogeny [130], a web service for phylogenetic analysis from sequence alignment to tree inference, and Phylemon [131], a suite of web tools for phylogenetics, phylogenomics, molecular evolution studies and hypothesis testing.’

Table 8: List of packages for phylogenetic analysis and evolutionary studies (*: tools that include web interface)

Software Features Link References

Bio++ Software package for sequence analyses, phylogenetic analysis, molecular evolution studies and population genetics analyses https://github.com/BioPP

[128,132]

CIPRES Server providing a software package for diverse phylogenetic analyses, including BEAST, FastTree, GARLI, IQ-TREE, jModelTest, MAFFT, MrBayes, PAUP, PhyloBayes and RAxML https://www.phylo.org/

[129]

HyPhy Software package for evolutionary analyses including evolution model selection, phylogenetic inference using ML and distance methods and sequence evolution studies https://www.hyphy.org/

[108]

NGPhylogeny* Workflows that integrate numerous methods and tools for multiple sequence alignment (MAFFT), trimming (BMGE), tree inference (FastTree, FastME or PhyML) and newick display

https://ngphylogeny.fr

[130]

Phylemon2* Web-tools for molecular evolution, phylogenetics, phylogenomics, molecular evolution studies and hypotheses testing. http://phylemon.bioinfo.cipf.es/index.html

[131]

Row 507-513: ‘Structure alignments can be realized to compare protein functions and evolution, and the mean distance in Å between homologous residues can be calculated. I-TASSER [159] and HHPred of the HH-suite software [160] can predict 3-dimentional structure for protein sequences using homology information. FoRSA [161] uses a structural alphabet known as Protein Blocks to identify a protein fold from its amino acid sequence, or to identify a protein sequence in the proteome of a species from a crystal structure by calculating a likelihood score.’

Table 10: List of programs for protein structure analyses

Software Features Link References

FoRSA Protein structure prediction using a structural alphabet http://www.bo-protscience.fr/forsa/

[161]

HHPred Protein structure prediction using homology information https://toolkit.tuebingen.mpg.de/tools/hhpred

[160]

I-TASSER Protein structure prediction and structure-based function annotation https://zhanglab.dcmb.med.umich.edu/I-TASSER/

[159]

PyMOL 3D visualization of molecules and diverse analyses on protein structures https://pymol.org/2/

[158]

FigTree should not belong to the list of nucleic acid databases (table 1).

We removed FigTree from Table 1

1. Link for UniProt in table 2 should not be directed to a specific protein (P04637) but be generalized.

We changed the link to the generalized web page.

3. Gblocks Server is no longer available (link in table 4).

The broken link was replaced with the valid one.

4. Links for BayesTraits, HGT-Finder in table 6 and Bio++ in table 7 do not work (tested on May 15th).

The links were replaced with valid ones.

5. Link for PyMol in table 8 is directed to Mesquite.

The link was replaced with a link to PyMol

6. The reference for some tools are missing in the main text and tables. Namely, ClustalOmega, Probcons (table 4), BayesTraits, FigTree, PAUL (table 6), and Bio++ Suite (table 7).

References were added to each of these tools.

Reviewer 2:

In general, I think there is merit in publishing a manuscript like the one submitted. However, I also think you could make it more comprehensive and more accurate, especially regarding the section on the protocol and phylogenetic analysis. Regarding the protocol, I think readers would need to have a more detail decision tree that offers them alternative paths, depending on what their objectives and data are. You need to incorporate some measures of quality control at different steps in the protocol, and you need feedback loops that are followed in case the result of a quality control is that the preceding analysis did not yield the expected result (otherwise you will perpetuate errors made at the early steps). In this regard, your protocol is a little like that in Ciccarelli et al. (2006; Science 311, 1283-1287).

We are grateful that the reviewer also sees the merit in this practical guide to the study of gene and protein phylogeny and evolution for those willing to get into the field. This first comment is certainly also valid. Therefore, we have edited the text so that an extra step to assess phylogenetic assumptions after the alignment trimming, and several tools are present in the protocol. We added feedback loops in the protocol (Figure 1).

Row 88-91: ‘Feedback loops illustrate the necessity to control the quality of the alignment, to assess phylogenetic assumptions and to test the robustness of the tree, and to go back to previous steps to redo the analysis if necessary.’

We also added several tools proposed by Reviewer 2. This makes the manuscript more comprehensive, especially for sequence alignment and trimming, model selection and phylogenetic analysis. We also edited the text to clarify the specificities of numerous tools and methods so users can select the best fit method depending on their data and objectives, depending on the size of their datasets. More specifically, we have edited at the following paragraphs:

Row 248-256 : ‘Once the alignment is completed, it is necessary to select the positions and regions that will be used for the phylogenetic inference. Poorly aligned positions and highly variable regions are not phylogenetically informative, because these positions might not be homologous or subject to saturation. These positions should be manually or automatically excluded prior to the phylogenetic analysis. The resulting sub-MSA maximizes the phylogenetic signal of the alignment [51]. Alignment trimming can be done manually or using appropriate programs. The completeness of alignments can be quantified and phylogenetic informative regions of the alignment can be selected using appropriate tools, such as Guidance 2 [52], AliStat [53], Gblocks [51,54], trimAl [55], BMGE [56] and Noisy [57] (Table 5).’

We also included more steps to the protocol, including a paragraph on the validation of phylogenetic assumptions, based on a new tool that has been implemented in IQ-TREE, and another paragraph on bootstrapping methods.

Row 261-269: ‘Most phylogenetic methods rely on simplifying assumptions stating for example that all sites in the alignment evolved under the same tree, that mutation rates have remained constant, and that substitutions are reversible. Once the alignment is performed and the sites selected for phylogenetic inference, a recent phylogenetic protocol recommends assessing those phylogenetic assumptions when possible [58]. If the phylogenetic data violate these assumptions, the phylogeny and evolutionary analyses can be biased with most common phylogenetic programs [59]. Several statistical methods have been developed. Recently, tests for all these assumptions have been included in IQ-TREE [60,61]. It is also possible to use the R package MOTMOT [62].’

I welcome that you cite a lot of databases, but could you move their citations from where they are to right after the name? Sometimes, a citation is at the end of the sentence and might be mistaken for a citation to something else (e.g., other databases or multiple sequence alignment).

Yes, thanks for pointing this out. Now, we have edited through all the manuscript the text to consistently have citations right after the name of every database and for software every time they appear in a paragraph. 

Regarding the phylogenetic analysis (L194-L364), I compliment you on a valiant attempt to cover a large and complex field. However, I do not think you succeeded because I found gaping holes:

1. You seem unaware of previous phylogenetic protocols, one of which appeared recently in NAR Genomics & Bioinformatics (2, lqaa041).

Thanks for pointing this paper out to us. Now, we have edited figure 1 and the text according to this recently published protocol, and cite this work. We have edited the text on:

Row 264-265: ‘a recent phylogenetic protocol recommends assessing those phylogenetic assumptions when possible [58]’

Row 370-373: ‘For model-based methods, a recent phylogenetic protocol recommends to test the goodness of fit between tree, model and data using a parametric bootstrap [58,128]. Bayesian inference method calculates posterior probabilities, which measure branch support instead of bootstrap values.’

2. Because the manuscript does not present something novel but summarizes bioinformatics tools and resources, chiefly databases, you need to be comprehensive. Unfortunately, you were not comprehensive. This applies to both multiple sequence alignment methods and phylogenetic methods.

We have edited the text to highlight that the main aim of this work is to provide a roadmap for the study of gene and protein evolution for the broad scientific community and scientist that are new to the field. Our intention is indeed to be comprehensive, but also simplify the decision-making process for the reader while choosing the most appropriate tools for their scientific endeavor. Therefore, we believe it is important to have a balance between being exhaustive and practical enough in a way that will also incorporate beginners. This paper does not aim to present all tools that exist for MSA or phylogenetic analysis but rather a set of tools which are diverse, widely used in the scientific community and also user friendly. Specially for beginners in the field, it is easier to navigate a list of software that is actively maintained by the scientific community, and for which getting support (in the form of tutorials, documentations, publications, online support groups) is easier. Still, we added several tools and methods for sequence alignment, trimming, model selection and phylogenetic inference (see responses below). We have edited text:

Row 5: ‘a practical guide’

Row 69-70: ‘The aim is to provide a practical guide for beginners and more advanced explorers into protein and gene phylogeny and evolution.’

3. Currently, I am comparing the accuracy of 20 multiple sequence alignment methods (i.e., Clustal Omega, CONTRAlign, DIALIGN-TX, Fsa, GramAlign, KAlign, MAFTT (default), MAFFT (EINSI), MAFFT (GINSI), MAFFT (LINSI), MSAProbs, Muscle, Pasta, PicXAA, Poa, Prank, ProbAlign, ProbCons, T-Coffee (fast), and T-Coffee (regressive)). You considered only a few of these.

Thanks for pointing out also these methods. Related to our response in the previous point, we recognize that there are several other tools that have been developed. We have therefore added a majority of these methods in the text and in Table 4. 

Thus, we have edited the following: 

Row 226-243: ‘The progressive approach aligns progressively from the closest to the most distant sequences. It is used by CLUSTALW [34], CLUSTAL Omega [35], MUSCLE [36], PRANK [37,55], Kalign [39] and MAFFT [40]. Consistency-based methods calculate the best alignment after different pairwise alignments. They are used by T-COFFEE [41] and PROBCONS [42] and its successor CONTRAlign [43], the latter for amino acid sequences only (Table 4). CLUSTALW [34] and MUSCLE [36] are included in MEGA [44]. They display web interfaces, as well as MAFFT [40], Kalign [39], and PRANKS [37,55]. PROBCONS [42], T-COFFEE [41] and MAFFT are described to have particularly high accuracy but also high calculation times [45]. They should be restricted to small and intermediate datasets. CLUSTAL Omega [35] and Kalign [39] are particularly fast, but less accurate [46]. They can be used for datasets of up to 4000 and 2000 sequences, respectively [45,46]. The performances of Muscle are intermediate [46]. PRANK is particularly accurate for large sets and closely related sequences. Bali-Phy [47] performs a bayesian co-estimation of alignment, phylogeny, and other parameters and is also argued to be very reliable. PASTA [48] and UPP [49], that uses a machine learning technique, are designed for very large datasets. MAFFT offers a wide range of methods, which can be accuracy-oriented, such as L-INS-i, G-INS-I and E-INS-i; or speed-oriented, such as FFT-NS-2, which can be used for up to 30 000 sequences.’

Table 4: List of programs for multiple sequence alignment (*: tools that include web interface)

Software Features Link References

BAli-Phy Multiple sequence alignment of nucleotide and amino acid sequences and phylogenetic analysis using a bayesian approach http://www.bali-phy.org

[47]

CLUSTAL Omega* Speed-oriented multiple sequence alignment for nucleotide or amino acid data, suitable for large datasets https://www.ebi.ac.uk/Tools/msa/clustalo/

[35]

CLUSTALW* Multiple sequence alignment for nucleotide or amino acid data https://www.genome.jp/tools-bin/clustalw

[34]

CONTRAlign (ProbCons) Accuracy-oriented multiple sequence alignment for amino acid data http://contra.stanford.edu/contralign/

[42,43]

Kalign* Multiple sequence alignment for nucleotide or amino acid data, suitable for large datasets https://www.ebi.ac.uk/Tools/msa/kalign/

[39]

MAFFT* Accuracy-oriented multiple sequence alignment for nucleotide or amino acid data https://mafft.cbrc.jp/alignment/server/

[40]

MUSCLE* Multiple sequence alignment for nucleotide or amino acid data https://www.ebi.ac.uk/Tools/msa/muscle/

[36]

PASTA Speed-oriented multiple sequence alignment for nucleotide or amino acid data, designed for very large datasets https://bioinformaticshome.com/tools/msa/descriptions/PASTA.html

[48]

PRANK/

WebPRANK* Speed-oriented multiple sequence alignment for nucleotide or amino acid data, should be preferred for close sequences and large datasets http://wasabiapp.org/software/prank/
https://www.ebi.ac.uk/goldman-srv/webprank/

[37,38]

SATé Software package for multiple sequence alignments and phylogenetic inference https ://phylo.bio.ku.edu/software/sate/sate.html

[50]

T-COFFEE* Accuracy-oriented multiple sequence alignment of nucleotide and amino acid sequences http://tcoffee.crg.cat/

[41]

UPP Speed-oriented multiple sequence alignment of nucleotide and amino acid sequences, designed for very large data sets https://github.com/smirarab/sepp.

 [49]

However, several of the tools mentioned by reviewer #2 are not maintained, or have not been updated in several years. Sometimes they are not accessible (links in the publications are broken, do not exist and so one needs to contact the authors to get the source code). We recognize the value of these tools and their contribution to the field, but we believe including them in our manuscript may only create confusion. For those reasons, we have not included Probalign, PicXAA, FSA, GramAlign, and MSAProbs.

4. You mention one method for trimming sites from multiple sequence alignment (GBlocks; in Table 4). There is a suite of other and more suitable methods available (see NAR Genomics & Bioinformatics 2, lqaa024; see also citations 13-21 in that paper).

Yes, we thank the reviewer for this thought. We added a full paragraph on alignment trimming and mention several tools from these publications.

Row 248-256: ‘Once the alignment is completed, it is necessary to select the positions and regions that will be used for the phylogenetic inference. Poorly aligned positions and highly variable regions are not phylogenetically informative, because these positions might not be homologous or subject to saturation. These positions should be manually or automatically excluded prior to the phylogenetic analysis. The resulting sub-MSA maximizes the phylogenetic signal of the alignment [51]. Alignment trimming can be done manually or using appropriate programs. The completeness of alignments can be quantified and phylogenetic informative regions of the alignment can be selected using appropriate tools, such as Guidance 2 [52], AliStat [53], Gblocks [51,54], trimAl [55], BMGE [56] and Noisy [57] (Table 5).’

5. You mention three model-selection methods but overlooked other more flexible methods (Nature Methods 14, 587-589; Mol. Biol. Evol. 29, 1695-1701; Syst. Biol. 63, 726-742; Mol. Biol. Evol. 34, 772-773; Syst. Biol. 69, 249-264). 

Yes, we thank the reviewer also for pointing this out. We included several new tools in the text and in Table 4 (see response above). More specifically, the edits are made in

Row 279-281: ‘More recently, the GHOST model for alignments with variation in mutation rate was introduced and implemented in IQ-TREE.’

Row 288-292: ‘PartitionFinder 2 [82] can be used with nucleotide and amino acid data. Model test selectors are also included in programs such as MEGA [44] and PhyML (SMS) [83]. ModelFinder [84] is a model selection method for alignments or nucleotides, codons or amino acids implemented in IQ-TREE [60,61], that includes a flexible model of rate-heterogeneity between sites.’

Table 6: List of programs for molecular evolution model selection

Software Features Link References

ModelFinder Fast model selection with a model of rate heterogeneity between sites (nucleotide, amino acids or codons) Implemented in IQ-TREE [84]

ModelTest / jModelTest Nucleotide substitution model selection http://evomics.org/resources/software/molecular-evolution-software/modeltest/

[80]

PartitionFinder 2 Molecular evolution model selection (nucleotide or amino acids) http://www.robertlanfear.com/partitionfinder/

[82]

ProtTest Aminoacid substitution model selection https://github.com/ddarriba/prottest3

[81]

SMS Molecular evolution model selection included in PhyML (nucleotide or aminoacid) http://www.atgc-montpellier.fr/sms/

[83] 

6. Your understanding of the relationship between log-likelihood and the AIC and BIC is wrong (L232-L233), suggesting confusion. You should read Briefings in Bioinformatics (21, 533-565).

We have now edited this sentence to make it more accurate and clear.

Row 283-284: ‘For every substitution model, these tools calculate the Bayesian information criterion (BIC) [78] and the Akaike information criterion (AIC) [79] from the log-likelihood scores.’

7. Your understanding of what the non-parametric bootstrap scores indicate is wrong (see Science 310, 1911-1912).

We acknowledge that the text can be much clearer. Therefore, we have added a full paragraph with information, precision and references on bootstrapping method and other tests of tree robustness. See edits on

Row 355-373: ‘Once the phylogenetic tree is obtained, it is recommended to estimate the robustness of the nodes. Most programs of phylogenetic analysis use the non-parametric bootstrapping method [121]. Bootstrapping is an estimate of error used to assess the repeatability of the clade and the how consistently the data support the nodes [121,122]. The characters (e.g., nucleotides or amino acids) are randomly resampled with replacement and a new phylogeny is calculated for each replicate. A bootstrap value is calculated for every node, indicating the proportion of replicate phylogenies that recovered the node from the initial tree. A bootstrap value of 100% means that the node is supported by all informative characters, while low values mean that only few characters support the node. A bootstrap value above 95% is usually considered very good and a bootstrap value below 75% is generally considered a poor support for the clade. 1000 replicates are often used in phylogenetic analysis. Since bootstrapping can be time consuming, fast approximation methods for phylogenetic bootstrap have been proposed and are implemented in programs such as RAxML or IQ-TREE [123–125]. Other tests of branch support robustness exist, such as the Shimodaira-Hasegawa test [126] and the approximate likelihood ratio test [127], both implemented in PhyML along with the bootstrapping method. FastTree includes only the Shimodaira-Hasegawa test. For model-based methods, a recent phylogenetic protocol recommends to test the goodness of fit between tree, model and data using a parametric bootstrap, rather than non-parametric bootstrap [58,128]. Bayesian inference method calculates posterior probabilities, which measure branch support instead of bootstrap values.’

8. You mention several distance matrix-based phylogenetic methods but do not cite papers describing them. 

To clarify, we have complemented citations in the text. Edits are made on

Row 314-315: ‘Distance methods include the Unweighted Pair Group Method with Arithmetic mean (UPGMA) [91], Neighbor Joining (NJ) [92], and Minimum Evolution (ME) [93].’

9. Your list of phylogenetic methods is incomplete and misses many important programs (e.g., IQTREE (Mol. Biol. Evol. 32, 268-274), IQTREE2 (Mol. Biol. Evol. 37, 1530-1534), PHYLIP (Felsenstein, 2005), PyCogent (Genome Biology 8, R171), FastME v2.0 (Mol. Biol. Evol. 32, 2798-2800), LSD (Syst. Biol. 65, 82-97) Garli (Syst. Biol. 63, 812-818), PhyloBayes 3 (Bioinformatics 25, 2286-2288), SplitsTree (Mol. Biol. Evol. 23, 254-267)).

Yes, the reviewer is correct that not all programs are presented. This is, as mentioned above, a deliberation to balance the overview with a guide that is accessible also to novices to the field. For phylogenetic analysis tools we have the same criteria as for the MSA (not include tools which are not maintained or are difficult to access). Therefore, we added most of these tools to be more comprehensive, with comments on their specificities. This concerns particularly ML methods and Bayesian inference. (see also responses to comments 2-4, and 9). More specifically, the following is edited:

Row 329-339: ‘Programs for ML phylogenetic analysis include MEGA [44], SeaView [104], PhyML [105], RAxML [106], FastTree [100], PAML [107], PAUP [99,103], IQ-TREE [60,61], HYPHY [108], PHYLIP [101] and GARLI [109,110] (Table 7). All of them can be used with nucleotide or amino acid data. MEGA [44] and SeaView [104] are known to be very user-friendly. They include sequence alignment tools and tree editors. PhyML [105] is accurate, easy of use and, like PAUP [103] and MEGA [44], includes all common models of molecular evolution. RAxML [106] and particularly FastTree [100] are fast and well suited for large datasets (up to 1 million sequences with FastTree). They use a specific model of rate heterogeneity, in addition to Gamma law and proportion of invariant sites. IQ-TREE [60,61], that includes ModelFinder [84] and a very fast bootstrapping method, is reported to be both fast and accurate [111]. PAUP [99,103] is slower than other programs, and uses nucleotide data only.’

Row 343-349: ‘The most recent method for phylogenetic reconstruction uses Bayesian inference, that calculates the probability of the molecular evolution model given the data. The main software used for BI-based phylogenetics is MrBayes [112] that uses the Markov Chain Monte Carlo (MCMC) algorithm (Table 7). PhyloBayes [113,114] is a bayesian MCMC sampler for phylogenetic reconstruction with protein data using a specific probabilistic model, well adapted for large datasets and phylogenomics. Bali-Phy [47] can also be used for phylogenetic analysis using Bayesian inference.’

Table 7: List of programs for phylogenetic analysis using distance methods, maximum parsimony, maximum lilekihood (ML) and bayesian inference (*: tools that include web interface)

Software Features Link References

APE R-written package for molecular phylogenetics http://ape-package.ird.fr

[98]

BAli-Phy Sequence alignment and phylogenetic inference using a bayesian approach http://www.bali-phy.org

[47]

BayesTraits Phylogenetic inference and other evolutionary analyses using Bayesian inference http://www.evolution.reading.ac.uk/BayesTraitsV4.0.0/BayesTraitsV4.0.0.html

[115]

ETE Toolkit Visualization and analysis of phylogenetic trees http://etetoolkit.org/

[116]

FastMe Fast phylogenetic inference using distance methods. http ://www.atgc-montpellier.fr/fastme/

[94]

FastTree Phylogenetic inference using ML for nucleotide (GTR and JC models) and amino acid (JTT and WAG models), and Shimodaira-Hasegawa test. Suitable for very large datasets. http://www.microbesonline.org/fasttree/

[100]

FigTree Graphic software for phylogenetic trees http://tree.bio.ed.ac.uk/software/figtree/

[117]

GARLI Phylogenetic inference using ML for nucleotide (GTR model), aminoacid (most models) or codon data, with Gamma law and proportion of invariant sites. http://evomics.org/resources/software/molecular-evolution-software/garli/

[110]

HYPHY* Diverse evolutionary analyses including evolution model selection, phylogenetic inference using ML and distance methods and sequence evolution studies https://www.hyphy.org/

[108]

IQ-TREE ML phylogenetic inference, including model selection and ultrafast bootstrapping method. Includes the GHOST evolution model and tests for phylogenetic assumptions. http://www.iqtree.org/

[60,61]

ITOL Visualization and annotation of phylogenetic trees https://itol.embl.de/

[118]

MEGA Sequence alignment, model selection, phylogenetic analysis (parsimony, distance methods). Includes all common nucleotide and amino acid evolution models, Gamma law and proportion of invariant sites, and Bootstrapping method. https://www.megasoftware.net/

[44]

MrBayes Bayesian phylogenetic inference, ancestral states reconstruction, phylogenetic calibration and other evolutionary analyses http://nbisweden.github.io/MrBayes/

[112]

PAML Maximum likelihood phylogenetic inference, estimation of selection strength, ancestral states reconstruction and other analyses http://abacus.gene.ucl.ac.uk/software/paml.html

[107]

PAUP Phylogenetic inference using maximum parsimony and ML on nucleotide sequences (all ModelTest models), with Gamma law and proportion of invariant sites and Bootstrapping method. http://paup.phylosolutions.com/

[99]

PHYLIP Phylogenetic inference using parsimony, distance methods and ML https ://evolution.genetics.washington.edu/phylip.html

[101]

PhyloBayes Phylogenetic inference using Bayesian inference on proteins using a specific probabilistic model http ://www.atgc-montpellier.fr/phylobayes/

[113]

PhyML* Phylogenetic inference using ML, ancestral states reconstruction and various evolutionary analyses. Includes all common DNA and protein evolution models and diverse branch support methods (Bootstrap, Shimodaira-Hasegawa, aLTR…). https://github.com/stephaneguindon/phyml

Web interface : http://atgc.lirmm.fr/phyml/

[105]

PyCogent Phylogenetic inference and phylogeny drawing, various evolutionary analyses including partition models and ancestral states reconstruction https://github.com/pycogent/pycogent

[95]

RAxML Phylogenetic inference using ML with nucleotide (GTR) or amino acid data (all common models) with Gamma law or CAT and proportion of invariant sites. Suitable for large datasets. https://cme.h-its.org/exelixis/web/software/raxml/

[106]

SeaView Sequence alignment and phylogenetic inference using maximum parsimony, NJ and ML http://doua.prabi.fr/software/seaview

[104]

SplitsTree Phylogenetic inference, in particular unrooted trees, or phylogenetic networks https ://uni-tuebingen.de/fakultaeten/mathematisch-naturwissenschaftliche-fakultaet/fachbereiche/informatik/lehrstuehle/algorithms-in-bioinformatics/software/splitstree/

[97]

10. You really need to ensure that software and methods referred to are cited properly, preferentially every time and with version numbers included.

Yes, we have now been more careful on this. See response above (comments 8 and 10). In essence, we have edited the text to include the historical papers describing the different distance methods, as well as all software and methods.

11. Your figures are unclear, and the colours used are not consistent with a colour palette fit for colourblind people.

We thank the reviewer for pointing this out. Although most colors in our figures are defined by others (webtools and logos), we have changed the red color for yellow in Figure 3.

SPECIFIC COMMENTS

We thank the reviewer for the detailed comments below. To all, we have edited sentences to clarify and increase accuracy as suggested by the reviewers. Specific replies follow below (row numbers refer to those in the originally submitted MS).

L47-L48. Why start with “Although … community”?

Thanks. We removed “Although … community”.

L61. “allowed” > “has allowed”

Yes, thanks. We replace « allowed » with « has allowed ».

L61-L62. “nucleic acid and protein” > “nucleotides and amino acids” [DNA is a sequence of nucleotides and protein is a sequences of amino acids; therefore, it is not correct to write DNA sequences or protein sequences]

True. We replaced « nucleic acids » and « protein » with « nucleotide and amino acids ».

L62. “In addition, this data is” > “Typically, these data are” [datum is singular of data]

Yes, we changed « is » to « are ».

L66-L67. “navigate these resources can be puzzling” > “navigate through these resources can be challenging”

We changed « puzzling » for « challenging ».

L81. Delete “considerable accompanying”

We deleted « considerable accompanying ».

L84-L86. The figure is not a fully developed protocol like those in NAR Genomics & Bioinformatics 2, lqaa041 [2020].

We have updated the figure and added steps, in accordance with this protocol. We added alignment trimming, validation of phylogenetic assumptions, and bootstrapping as specific tools.

L89. “DNA sequences” > “nucleotide sequences (DNA)” [I am pedantic here, but DNA is a sequence (of nucleotides) so writing DNA sequences is the same as saying a sequence of a sequence of nucleotides]

We changed « DNA sequences » to « nucleotide sequences ».

L89. “and their protein translations” > “and, where applicable, their translations into protein”

We changed « and their protein translations » to « and, where applicable, their trabslations into protein ».

L103. FigTree is not a database!!! It is a phylogenetic tool, so it should be in a different table.

The FigTree is moved to the appropriate table.

L119. “protein sequences” > “amino-acid sequences” [I use a hyphen because amino and acid form a compound adjective]

We changed « protein sequences » into « amino-acid sequences ».

L123. See above.

We changed « protein sequences » into « amino-acid sequences ».

L148. “sequences alignment” > “pairwise sequence alignment” OR “multiple sequence alignment”

We changed “sequences alignment” to “pairwise sequence alignment or multiple sequence alignment”.

L159. “evolutionary relationships” > “evolutionary relationships (see below)”

We added « see below ».

L167 “under FASTA” > “in FASTA”

We changed « under FASTA » to « in FASTA ».

L177. “, i.e., … ancestry” > “(i.e., … ancestry)” [I prefer “that is,” in the sentence and “i.e.,” inside parentheses]

Parentheses are added.

L177-L179. This sentence should be so that the terms homology, orthology, paralogy and xenology are clear and unambiguous.

We rephrased the paragraph to explain more clearly the concepts of homology, orthology, paralogy and xenology.

L184. Include “(define/explain the E-value)”

Thanks, we added a definition of the E-value.

L185. “amino acid sequences” > “amino-acid sequences”

We added the hyphen.

L186. Explain “HH”

We introduced the HHsuite.

L196. “proteins within” > “proteins, even within”

We changed “proteins within” into “proteins, even within”.

L197. “data, (“ > “data (“

The comma is removed.

L198. “), includes” > “) includes”

The comma is removed.

L202. Delete “sequences”

We deleted « sequences ».

L203. “one after the other” > “, one after the other,”

Two commas were added.

L213-L214. Not “distantly related”; “closely related” [I spoke to Ary and Nick about this] Edited from “distantly related” to “closely related”.

L227. You list the two least realistic models of rate heterogeneity across sites. More realistic models of rate heterogeneity across sites are described in Kalyaanamoorthy et al. (Nature Methods 14, 587-589) and Crotty et al. (Syst. Biol. 69, 249-264).

We introduced these models and added the references.

L232-L233. This sentence is incorrect. The AIC and BIC are computed from the log-likelihood scores; not the other way around, as you write.

We rewrote the sentence to clarify.

Row 279-280: ‘For every substitution model, these tools calculate the Bayesian information criterion (BIC) [76] and the Akaike information criterion (AIC) [77] from the log-likelihood scores.’

L235-L239. Your description of the reason for doing a non-parametric bootstrap analysis is misleading and is not supported by relevant literature (e.g., Mol. Biol. Evol. 9, 366-369; Mol. Biol. Evol. 30, 1188-1195; Mol. Biol. Evol. 35, 518-522).

We added a paragraph to explain the bootstrapping and other methods to test tree robustness. We added the references.

L247-L250. Include citations after each method or program mentioned, please.

We added the citation immediately after every method and program described.

---

## [Decision Letter · Decision Letter 1]

6 Oct 2022

PONE-D-22-06715R1Roadmap to the study of gene and protein phylogeny and evolution – a practical guidePLOS ONE

Dear Dr. Hammarlund,

Thank you for submitting your manuscript to PLOS ONE. After careful consideration, we feel that it has merit but does not fully meet PLOS ONE’s publication criteria as it currently stands. Therefore, we invite you to submit a revised version of the manuscript that addresses the points raised during the review process.

We look forward to receiving your revised manuscript.

Kind regards,

Arndt von Haeseler

Academic Editor

PLOS ONE

Reviewers' comments:

Reviewer's Responses to Questions

**Comments to the Author**

1. Does the manuscript report a protocol which is of utility to the research community and adds value to the published literature?

Reviewer #1: Yes

Reviewer #2: Yes

2. Has the protocol been described in sufficient detail?

Descriptions of methods and reagents contained in the step-by-step protocol should be reported in sufficient detail for another researcher to reproduce all experiments and analyses. The protocol should describe the appropriate controls, sample sizes and replication needed to ensure that the data are robust and reproducible.

Reviewer #1: Yes

3. Does the protocol describe a validated method?

Reviewer #1: Yes

Reviewer #2: Yes

4. If the manuscript contains new data, have the authors made this data fully available?

Reviewer #1: N/A

Reviewer #2: No

**5. Is the article presented in an intelligible fashion and written in standard English?**

Reviewer #1: Yes

Reviewer #2: **No: **The manuscript requires further editing to improve clarity

6. Review Comments to the Author

Reviewer #1: The authors have adequately addressed the most important issues in this revised manuscript. I have no further comment.

Reviewer #2: GENERAL COMMENTS

In general, I still think there is merit in publishing a manuscript like the one submitted. However, I also think the manuscript can improved further. Several aspects of the revision suggest you were in a hurry to finish it, leading, in some cases, to confusion rather than clarity.

If you want the manuscript to become the paper that educators recommend as an introduction to the study of gene and protein phylogeny and evolution, then you may need to invest a bit more of your time on achieving an accurate, easy-to-read presentation of the topics. There is also evidence that you do not understand some of the aspects that you cover. Those have to be addressed so that you present a sound and reliable protocol, with references to critical databases, alignment tools, and phylogenetic methods.

In the following, I list some of the major issues:

1. An example of the above-mentioned misunderstanding appears in Figure 1: The feedback loops should be distinguished from the forward arrows (use different colour or dashed arrows). There are no arrows between “Select phylogenetic method” and the two boxes below it with methods.

2. Distance methods are model-based, so they should be listed with the maximum-likelihood and Bayesian methods.

3. You write “Test of robustness”, which is misleading. Robustness refers to the quality or condition of being strong and in good condition. A phylogenetic estimate can be robust, yet it may be incorrect. We would like the estimates to be accurate and the methods to be precise and accurate. As for the accuracy, we may test the goodness of fit between the tree, the model of sequence evolution, and the data. A good fit means that the tree and model of sequence evolution provide a good explanation of the data. Yet, we will never know whether that explanation is the correct (others may exist). A poor fit means that something is not right about the tree, the model of sequence evolution, or both. This is fundamental and it should be clear from your manuscript (which it isn’t).

4. The sections on alignment and phylogenetic analysis (L212-L395) have been improved a lot. However, you appear to ignore the new phylogenetic protocol [1], which you cited in the Introduction. I think it would serve you better if you described the different steps and procedures in the context of the new phylogenetic protocol [1] and/or Figure 1. You know the details and order of actions but newcomers to the field do not and would need a clear framework. The new protocol provides that framework.

5. Regarding multiple sequence alignment, I think you need to consider and cite some of the excellent papers by Morrison (e.g., Is Sequence Alignment and Art or Science; Syst. Bot. 40, 14-26 [2015]). You gloss over the challenges and pitfalls, some of which are illustrated in Golubchik et al. (Mol. Biol. Evol. 24, 2433–2442 [2007]), thereby belittling the challenge it is to obtain a trustworthy multiple sequence alignment. Apart from the progressive and consistency-based approach (mentioned on L224-L227), there are other ways to obtain a multiple sequence alignment. You should at least mention some of these other strategies, and cite the relevant literature (or reviews of that literature). Further, the consistency-based approach is not well described (L226-L227).

6. You should mention the minimum reporting standard for multiple sequence alignments (it was presented in Ref. 53) when introducing trimming of multiple sequence alignment. The reason for doing so is given in Ref. 53.

7. The section entitled “Test of robustness of phylogenetic trees” needs to be rewritten. For a start, choosing to use the word “robustness” is not good. Robustness usually refers to how accurate methods are when the assumptions of the methods are violated by the data (see Yang 2014; Molecular Evolution). Other words like accuracy, consistency and precision are more appropriate. In this context, note that:

• Accuracy means ‘the quality or state of being correct or precise’

• Consistency means ‘the quality of achieving a level of performance which does not vary greatly in quality over time’

• Precision means ‘the quality, condition, or fact of being exact and accurate’

Your description of bootstrapping is incorrect; it is not “an estimate of error” (L349). In my previous review, I drew your attention to the correct interpretation of bootstrapping, but you ignored that, which was unwise. Instead, you cite Hedges [120], which focuses on the number of bootstrap replicates needed gain an accurate estimate of the bootstrap P value, and not on the interpretation. You also need to note that bootstrap P values are both data and method dependent. It is the sites in the alignment, not “characters” (L350), which are sampled randomly with replacement, and the correct citation is missing: “each replicate” > “each replicate [Felsenstein 1985; Evolution 39, 783-791]”. Bootstrapping, as described by Felsenstein and implemented by many since then, is a non-parametric bootstrap approach. Using it gives us an insight into the consistency of the data (i.e., do we have enough sites in the data to consistently infer the same tree). It does not tell us whether the inferred tree is accurate.

You write that a “bootstrap value is calculated for every node” (L352). This is incorrect. The value is computed for every internal edge/branch/split that separates the sequences.

You write “from the initial tree” (L353). This is incorrect. It can be done for any tree, even the consensus tree (IQ-TREE often list the values for the ML tree and the consensus tree; and these trees may not be identical).

You write “A bootstrap value …for the clade”. Who did you quote? There has been some disagreement on the interpretation of bootstrap P values. New methods like those from von Haeleser’s group [121, 122] have clarified the matter considerably.

You write “1000 replicates …” (L356). This should be spelled out, and you should probably mention how that fits with Hedges [120].

You write “Other branch support methods exist” (L359) and then refer to the Shimodaira-Hasegawa test [124]. The bootstrap method is not a “branch support method”, so it is not correct to say “Other branch …”. Moreover, the Shimodaira-Hasegawa test is not a branch support test either. It is a test designed to compare the ML tree to other trees.

8. I have not included any specific comments on the two case studies, mainly because I have already used far more time on this review than I had available. That said, the two sections do need some revision.

9. None of the figures are as clear as is necessary for publication. In fact, I can’t read them.

The Bibliography is clearly set using a French application and do not appear to comply with PLoS One’s style. Please ensure that the Bibliography complies with the journal’s formatting style. The Bibliography should have been fixed after the first review!

SPECIFIC COMMENTS

L25: “Here a” > “Here we present a”

L26: “nucleic acid” > “DNA”

L27: Delete “is presented”

L33: “proto cells to modern organisms” > “primitive cells to modern cells and multicellular organisms, ”

L36: “sequence composition of” > “information content in”

L38-L40: Poor sentence structure

L41: “mutations that” > “mutations (including substitutions) that”

L43: “all” > “both”

L45: “past decades” > “past four decades”

L48: “taxa” > “taxa, genes or genomic components (e.g., transposable elements)”

L52: “the others” > “other species”

L53: “infection diseases” > “infectious diseases”

L59: Delete “both”

L65: Delete “the”

L66: “ and of” > “as well as”

L67: “processes, following” > “processes. In so doing, we follow”

L68: “used by many and maintained” > “maintained and used by many”

L75: Delete “peer-reviewed”

L80: Delete “digitalized

L82: “nucleic acid or amino acid sequence” > DNA or amino-acid sequences”

L88: Perhaps state that the protocol is based on that in [1].

L89: “Nucleic acid” > “DNA”

L95: ”browser focusing on chordates that contains” > ”browser that focuses on chordates and contains”

L97: “This” > “The”

L99” Delete “Overall,”

L105: “rendered” > “provided”

L114: “provides several” > “provides access to several”

L116: “under” > “in the”

L117: “nucleotide sequences” > “nucleotide”

L118: Delete “pasted to fasta file”

L129: “format and the” > “format. The”

L129: “interlinks” > “links”

L130: “allows to batch download the” > “facilitates batch downloads of”

L132: “databases” > “databases (such as …)”

L136: “The KEGG focuses” > “The KEGG database focuses”

L148-L156: You alternate between “3-dimensional” and “3D”. Be consistent

L149: Delete “the”

L153: Place “i.e., … family” between parentheses

L159: You write “based on …, or both”. Perhaps separate the databases accordingly

L163: You write “based on the presence or absence of … strands”. I think you need to elaborate a bit further on this. There is SCOPe classification system to consider but you also need to consider the work by Sun et al. (Current Protocols in Protein Science (2004) 17.1.1-17.1.189)

L164: “class” > “classification”

L177: “[16], that combines” > “[16]. It combines”

L186: Delete “whole”

L188: “hemoglobine” > “haemoglobin”

L189: “hemoglobine” > “haemoglobin”

L190: “myoglobine” > “myoglobin”

L197: Include a citation to the E-value. You might know it, but you are targeting beginners who can be assumed to be ignorant.

L206: “retrieving from” > “retrieving sequences from”

L213: “related to each other” > “related to each other in an evolutionary sense” [they can be related to each other on other senses]

L220: “row one” > row of a matrix, one”

L230-L231: Poor sentence structure and you probably mean “user interface”, rather than “web interfaces”

L233: “calculation times” > “execution times”

L236-L237: You need a reference to support this statement, and you then need to cite Golubchik et al. (Mol. Biol. Evol. 24, 2433–2442 [2007]), who reported the opposite for PRANK.

L238: “that” > “which”

L242: “web interface” > “user interface”

L253: “quantified and phylogenetic informative” > “quantified using AliStat [53] and phylogenetically informative”

L258: “stating for example that” > “stating, for example, that”

L259: “constant” > “constant over time (i.e., time-homogeneity)”

L260: “reversible” > “reversible and, therefore, also stationary (for details on these assumptions, see Jayaswal et al. (Syst. Biol. 63, 726-742 [2014]) and papers cited therein)”

L261: You write “can be biased [58]”. It would be more appropriate to cite Syst. Biol. (53, 623-637; 53, 638-643) because these papers used data generated by simulation (hence, the truth is known).

L262: “recent phylogenetic protocols recommend … possible [1]” > “a recent phylogenetic protocol [1] recommends … possible”

L264: “developped" > “developed”

L264: You write “tests for all these assumptions”. This is not true. Some have been included in IQ-TREE and IQ-TREE2. I agree that the matched-pairs tests of homogeneity (Bioinformatics 22, 1225-1231) are of relevance, but only for some Markovian conditions. One of these is implemented in Homo 2.1 (https://github.com/lsjermiin/Homo.v2.1).

L266: “Selection of the molecular evolution model” > “Selection of the optimal model of sequence evolution”

L267: Several papers and book chapters have described model selection in general terms. I think it would be wise to cite some of these (e.g., Front. Genet. 6, 319 [2015; Meth. Mol. Biol. 1525, 379-420 [2017]; DOI 10.1007/978-1-4939-6622-6_15).

L268: “Nucleotide or amino acid substitution exist” > “Several models of nucleotide or amino-acid substitution exist [REF]”. The statement requires some citations.

L270: you cite Yang 2006; you should cite is more recent book from 2014.

L273: You write “Each model … IQ-TREE [75]”. The set of models described here differ from the substitution models described in L271-L274. They are homotachous models of rate-heterogeneity across sites, so name them appropriately. You forgot to include the PDF/FreeRate model proposed by Yang (Genetics 139, 993-1005 [1995]) and now included in ModelFinder, IQ-TREE and PhyML.

L279: “every substitution model” > “model of sequence evolution (i.e., combination of substitution model and rate-heterogeneity across sites model)”

L281: “optimizing” > “minimising”

L284: Move “PartitionFinder 2 …[81]” to L288. The logic here is that ModelFinder is for a single partition and it supersedes ModelTest, jModelTest and ProtTest. PartitionFinder 2 is for multiple partitions.

L293: Replace the sentence with “Phylogenetic trees may consider the topology and the branch lengths (phylograms) or just the topology (cladograms)”

L294: “tree building methods” > “tree-building methods” [compound adjective]

L295: “distance” > “distances”

L295-L206: “number of differences” > “”numbers and types of differences”

L296: “to reconstruct” > “and they use this matrix to reconstruct”

L296-L297: The sentence is wrong. Distance methods are also character-based methods.

L299: “classic” > “classical”

L303: Include citation after “long branch attraction”

L309: “mean” > “Mean”

L310: “[91].” > “[91] tree-inference methods.”

L312: “amino acid data” > “amino-acid data” [compound adjective]

L315: “but also” > “and”

L316: Delete “,”

L317: “in R language that” > “in the R language. It”

L320: “ML” > “maximum likelihood [ML]”. I think this is the first time that you typed ML, so it has to be defined, even though you and I both know what it refers to

L322: “ML methods” > “Maximum-likelihood methods”. When starting a sentence, it is good practice to spell out the abbreviation (or number)

L322: “probability” > “likelihood”

L323: “ML aims” > “Maximum likelihood aims”

L323: “combinations of model” > “combinations of trees and model”

L327: “amino acid data” > “amino-acid data”

L328: “is accurate” > “is reported as being accurate”

L329: “all common models of molecular evolution” > “many common models of substitution” or “many common models of sequence evolution” [choose the most accurate statement”

L331: “rate heterogeneity … invariant sites” > “rate-heterogeneity across sites (…)”. Here, you replace “…” with the model of that you refer to

L333: “that” > “which”

L333: “method” > “method [REF]”. Here you need to replace “REF” with the relevant citation to the method implemented in IQ-TREE and IQ-TREE 2

L336: “recent method” > “recently-developed method”

L336: “inference, that” > “inference (BI). It”

L337: “calculates the probability” > “calculates the posterior probability”

L337: “the molecular evolution model given” > “the tree and model of sequence evolution, given”

L338: “[109] that” > “[109]. It”

L362: “inbcludes” > “includes”

L365: “taxa” > “sequences”

L366: “addressed question” > “question asked”

L370: “taxa” > “sequences”

L370: “studied ingroup” > “ingroup of interest”

L373: “identified, … viruses,” > “included”

L375: “of the longest branches” > “between the most dissimilar sequences in the tree”

L378: “exported using … a graphical” > “visualised using a graphical”

L394: You should include Geneious (https://www.geneious.com/) in the table and mention it in the text.

L397: “allows” > “allow users”

L404: Here you should only write MP, ML and BI. Note that you need to define MP somewhere earlier on in the text

L405: “molecular” > “sequence”

L416: In this section, you use the term “mutations”; typically the term “substitutions” is used

L417: “The strength” > “The type and strength”

L417: “can be calculated. This” > “may be of interest. It”

L418: “ration” > “ratio”

L421: “The ratio … 1, that” > “If dN/dS > 1, then the”

L426: “Phylogenetic calibration” > “Molecular dating of events”

L427: “event using events” > “events, using other events”

L440-L444: This can be stated more succinctly

L450: Why do you refer to Table 8 here?

L457: “species” > “traits”

L462-L466: This needs to be stated better

L477: “Population genetics often studies” > Population geneticists often study”

L483: “of” > “at”

L484: “studies, for a full review see” > “studies; for a full review, see”

L493: “in Å” > “in ångström”

L494: You write “can be calculated”. Which program?

L495: “3-dimentional” > “3D”

L496: “amino acid” > “amino-acid”

L498: Here I think you need to include Alpha fold (Nature 596, 583–589 [2021])

7. PLOS authors have the option to publish the peer review history of their article (what does this mean?). If published, this will include your full peer review and any attached files.

Reviewer #1: No

Reviewer #2: No

2. Has the protocol been described in sufficient detail?

To answer this question, please click the link to protocols.io in the Materials and Methods section of the manuscript (if a link has been provided) or consult the step-by-step protocol in the Supporting Information files.

The step-by-step protocol should contain sufficient detail for another researcher to be able to reproduce all experiments and analyses.

Reviewer #2: Partly

---

## [Author Response · Author response to Decision Letter 1]

25 Nov 2022

We thank the reviewers for their helpful and insightful comments. We have revised the manuscript and responded to all comments and concerns of Reviewer #2. We believe that this review and our revision has resulted in a much-improved contribution. Below, we reply to all the comments one by one. We refer to new intervals of text by their row number in the manuscript.

Reviewer #1: The authors have adequately addressed the most important issues in this revised manuscript. I have no further comment.

Reviewer #2: GENERAL COMMENTS

In general, I still think there is merit in publishing a manuscript like the one submitted. However, I also think the manuscript can improved further. Several aspects of the revision suggest you were in a hurry to finish it, leading, in some cases, to confusion rather than clarity.

If you want the manuscript to become the paper that educators recommend as an introduction to the study of gene and protein phylogeny and evolution, then you may need to invest a bit more of your time on achieving an accurate, easy-to-read presentation of the topics. There is also evidence that you do not understand some of the aspects that you cover. Those have to be addressed so that you present a sound and reliable protocol, with references to critical databases, alignment tools, and phylogenetic methods.

In the following, I list some of the major issues:

1. An example of the above-mentioned misunderstanding appears in Figure 1: The feedback loops should be distinguished from the forward arrows (use different colour or dashed arrows). There are no arrows between “Select phylogenetic method” and the two boxes below it with methods.

Response: We changed the feedback loops to dashed arrows in Figure 1. We added an arrow between “select phylogenetic method” and the methods below.

2. Distance methods are model-based, so they should be listed with the maximum-likelihood and Bayesian methods.

Response: We added the box “selection of molecular evolution model” under the distance methods.

3. You write “Test of robustness”, which is misleading. Robustness refers to the quality or condition of being strong and in good condition. A phylogenetic estimate can be robust, yet it may be incorrect. We would like the estimates to be accurate and the methods to be precise and accurate. As for the accuracy, we may test the goodness of fit between the tree, the model of sequence evolution, and the data. A good fit means that the tree and model of sequence evolution provide a good explanation of the data. Yet, we will never know whether that explanation is the correct (others may exist). A poor fit means that something is not right about the tree, the model of sequence evolution, or both. This is fundamental and it should be clear from your manuscript (which it isn’t).

Response: We changed “test of robustness” to “test of reliability” on Figure 1 and in the text. We explain more in detail what a good fit means and we stress in the manuscript that a good fit doesn`t mean that the tree is correct but is just an estimate of goodness of fit between the tree, the model and the data.:

L455-457: “A good fit means that the tree and the model of sequence evolution provide a good explanation of the data but doesn’t indicate if the tree in correct or not.”. 

4. The sections on alignment and phylogenetic analysis (L212-L395) have been improved a lot. However, you appear to ignore the new phylogenetic protocol [1], which you cited in the Introduction. I think it would serve you better if you described the different steps and procedures in the context of the new phylogenetic protocol [1] and/or Figure 1. You know the details and order of actions but newcomers to the field do not and would need a clear framework. The new protocol provides that framework.

Response: We detailed more the new phylogenetic protocol and added information on the two new steps from this protocol: “assessing phylogenetic assumptions” and “test of goodness of fit”. 

L295-305 “Phylogenetic models rely on simplifying assumptions stating, for example, that all sites in the alignment evolved under the same tree (treelikeness), that mutation rates have remained constant over time (i.e., time-homogeneity), and that substitutions are reversible and, therefore, also stationary (for details on these assumptions, see [72]). If the phylogenetic data violate these assumptions, the phylogeny and evolutionary analyses can be biased [73–75]. Once the alignment is performed and the sites selected for phylogenetic inference, a recent phylogenetic protocol recommends assessing those phylogenetic assumptions when possible [1]. Statistical methods allowing to test stationarity, homogeneity under certain markovian conditions, and treelikeness have been developed and included in IQ-TREE and IQ-TREE2 [76,77]. Homo2.1 [78] is designed for the analysis of compositional heterogeneity in sequence alignments. It is also possible to use the R package MOTMOT [79].” 

L453-467: “It is possible that the inferred optimal model of sequence evolution used for the phylogenetic analysis is inadequate. Once a phylogenetic tree has been inferred, it is recommended to test the goodness of fit (i.e., the adequacy) between the tree, the model and the data [1]. A good fit means that the tree and the model of sequence evolution provide a good explanation of the data but doesn’t indicate if the tree in correct or not. The goodness of fit can be tested using a parametric bootstrap [149]. Parametric bootstrap consists in using the optimal tree and the optimal model as an input to simulate sequence evolution to generate pseudo-data with a Monte Carlo simulation. Sequence generating data, such as SeqGen (https://github.com/rambaut/Seq-Gen/releases/tag/1.3.4, [150]) can be used. The goodness of fit is calculated from the difference between the unconstrained and constrained (i.e., assuming the optimal tree and the optimal model) log-likelihoods of the real data and the pseudo-data. If the fit is poor, it can be good to check the alignment and the selected set of sites and/or the sequence evolution model (feedback loops on Fig.1). Several methods have been developed to test the adequacy of the data [151]. The Goldman-Cox (GC) test can be used with several phylogenetic programs such as PAUP. Most Bayesian phylogenetic programs employ the posterior predictive (PP) test.” 

We stress the importance of testing phylogenetic assumptions prior to phylogenetic analysis and the goodness of fit. Our Figure 1 and our manuscript is now based on this new protocol.

5. Regarding multiple sequence alignment, I think you need to consider and cite some of the excellent papers by Morrison (e.g., Is Sequence Alignment and Art or Science; Syst. Bot. 40, 14-26 [2015]). You gloss over the challenges and pitfalls, some of which are illustrated in Golubchik et al. (Mol. Biol. Evol. 24, 2433–2442 [2007]), thereby belittling the challenge it is to obtain a trustworthy multiple sequence alignment. Apart from the progressive and consistency-based approach (mentioned on L224-L227), there are other ways to obtain a multiple sequence alignment. You should at least mention some of these other strategies and cite the relevant literature (or reviews of that literature). Further, the consistency-based approach is not well described (L226-L227).

Response: We added a paragraph to discuss more in detail the challenge of obtaining accurate MSA where we cite Morrison`s paper and Golubchik`s work. 

L258-277: “In practice, finding the accurate multiple sequence alignment can be challenging for several reasons. For example, computer programs are not based explicitly on the hypothesis of homology between aligned residues. They consider homology exclusively as similarity, and do not consider the other conditions for homology such as congruence and conjunction [61]. Furthermore, one should keep in mind that the alignment with the best score is not necessarily biologically correct. It is also difficult to get a good alignment for sequence that have diverged significantly and sharing low identity. In this case, for protein-coding sequences, amino-acid data should be preferred over nucleotide data, since it is possible, for example, to consider the biochemical similarity of substitutions of amino acids. Alignment software require defining a gap-opening penalty and a gap-extension penalty, but these values are arbitrary. It is common that different sequences in the alignment do not have the same length, for biological or experimental reasons. It is recommended to keep end gaps unpenalized [62]. Furthermore, indels are reported to affect the accuracy of MSA programs. It is recommended to use several MSA programs for sequences that contain indels [63]. MAFFT is reported to be the most accurate program in the case of sequence with non-overlapping deletions and in the case of alternatively spliced gene products [63]. Another difficulty concerns the case of repeated sequences with different numbers of repeats. Here, a domain of one sequence can be homologous to several domains of another sequence. Single nucleotides or small sequences can also be repeated like in the case of microsatellites, as well as entire protein domains. In the last case, it is recommended to excise the repeated domains [62].”

We describe more in detail consistency-based methods, and we cite the other MSA methods:

L242-244: “Other approaches include the iterative refinement method, which is also included in MAFFT and Muscle [51], the genetic algorithms, used by SAGA [52], and the methods using hidden Markov models, such as SAM [53].”.

6. You should mention the minimum reporting standard for multiple sequence alignments (it was presented in Ref. 53) when introducing trimming of multiple sequence alignment. The reason for doing so is given in Ref. 53.

Response: We added a mention to this minimum reporting standard:

L287-288: “A minimum reporting standard has been developed to quantify the alignment completeness, and implemented in AliStat [66].”

7. The section entitled “Test of robustness of phylogenetic trees” needs to be rewritten. For a start, choosing to use the word “robustness” is not good. Robustness usually refers to how accurate methods are when the assumptions of the methods are violated by the data (see Yang 2014; Molecular Evolution). Other words like accuracy, consistency and precision are more appropriate. In this context, note that:

• Accuracy means ‘the quality or state of being correct or precise’

• Consistency means ‘the quality of achieving a level of performance which does not vary greatly in quality over time’

• Precision means ‘the quality, condition, or fact of being exact and accurate’

Your description of bootstrapping is incorrect; it is not “an estimate of error” (L349). 

Response: We rewrote the part on non-parametric boostrap: 

L411-437“Once the phylogenetic tree is obtained, it is recommended to estimate the reliability of the clades. Most programs of phylogenetic analysis use the non-parametric bootstrapping method [142]. It is a resampling technique used to assess the repeatability of the clade and the how consistently the data support the nodes [143]. The sites in the alignment (e.g., nucleotides or amino acids) are randomly resampled with replacement and a new phylogeny is calculated for each replicate [142]. A bootstrap value is calculated for every internal branch, indicating the proportion of replicate phylogenies that recovered the clade. A bootstrap value of 100% means that the node is supported by all informative characters, while low values mean that only few characters support the node. Bootstrapping gives a measure of the consistency of the estimate, but it is not a measure of the accuracy of the tree [144]. The number of replicates that are necessary to obtain a good accuracy of the bootstrap value depends on the bootstrap value. For example, for a 1% confidence interval on a bootstrap value of 95, 2000 replicates are necessary [143].

Since bootstrapping can be time consuming, UFBoot and UFBoot2, fast approximation methods for phylogenetic bootstrap, have been developed and are implemented in programs such as RAxML or IQ-TREE [130,145,146]. They are also less biased than other non-parametric bootstrapping methods and robust against moderate model violations. While other bootstrapping methods tend to underestimate the probabilities of the clade of being correct, the bootstrap values from UFBoot and UFBoot2 truly reflect the probability of the clade of being correct, simplifying the interpretation of Bootstrap values (a bootstrap value of 95% indicate a probability of 95% to be correct) [145]).

With ML-based phylogenetic inference, the Shimodaira-Hasegawa test, and its improved version AU [147], are designed to evaluate alternative phylogenetic hypotheses, and test if a tree is better supported than another one. It can be used with PAUP, PhyML, FastTree and IQ-TREE. The approximate likelihood ratio test [148] is another test of phylogenetic relationship, implemented in PhyML along with the bootstrapping method. Bayesian inference method calculates posterior probabilities, which measure branch support instead of bootstrap values.”. 

We changed the title to “Test of reliability of phylogenetic trees”. We replaced “nodes” with “branches”. We also added a few comments on the interpretation of bootstrap, and how UFBoot provides a less biased bootstrap estimate, that facilitates the interpretation of bootstrap values. We changed “sites” into “characters”. We explain more in detail the non-parametric bootstrap.

In my previous review, I drew your attention to the correct interpretation of bootstrapping, but you ignored that, which was unwise. Instead, you cite Hedges [120], which focuses on the number of bootstrap replicates needed gain an accurate estimate of the bootstrap P value, and not on the interpretation. You also need to note that bootstrap P values are both data and method dependent. It is the sites in the alignment, not “characters” (L350), which are sampled randomly with replacement, and the correct citation is missing: “each replicate” > “each replicate [Felsenstein 1985; Evolution 39, 783-791]”. 

Response: We added a reference to Felsenstein`s paper on non-parametric bootstrap.

Bootstrapping, as described by Felsenstein and implemented by many since then, is a non-parametric bootstrap approach. Using it gives us an insight into the consistency of the data (i.e., do we have enough sites in the data to consistently infer the same tree). It does not tell us whether the inferred tree is accurate.

Response: We added a comment to stress that non-parametric boostrap is not a measure of the accuracy of the tree “Bootstrapping gives a measure of the consistency of the estimate, but it is not a measure of the accuracy of the tree.”

You write that a “bootstrap value is calculated for every node” (L352). This is incorrect. The value is computed for every internal edge/branch/split that separates the sequences.

You write “from the initial tree” (L353). This is incorrect. It can be done for any tree, even the consensus tree (IQ-TREE often list the values for the ML tree and the consensus tree; and these trees may not be identical).

Response: We corrected the sentence into “A bootstrap value is calculated for every internal branch, indicating the proportion of replicate phylogenies that recovered the clade.”

You write “A bootstrap value …for the clade”. Who did you quote? There has been some disagreement on the interpretation of bootstrap P values. New methods like those from von Haeleser’s group [121, 122] have clarified the matter considerably.

We added a comment on the interpretation of boostraps values and how the UFBoot method facilitates their interpretation. L424-431: “Since bootstrapping can be time consuming, UFBoot and UFBoot2, fast approximation methods for phylogenetic bootstrap, have been developed and are implemented in programs such as RAxML or IQ-TREE [130,145,146]. They are also less biased than other non-parametric bootstrapping methods and robust against moderate model violations. While other bootstrapping methods tend to underestimate the probabilities of the clade of being correct, the bootstrap values from UFBoot and UFBoot2 truly reflect the probability of the clade of being correct, simplifying the interpretation of Bootstrap values (a bootstrap value of 95% indicate a probability of 95% to be correct) [145]).”

You write “1000 replicates …” (L356). This should be spelled out, and you should probably mention how that fits with Hedges [120].

Response: We added comments on the number of bootstrap according to Hedges, “The number of replicates that are necessary to obtain a good accuracy of the bootstrap value depends on the bootstrap value. For example, for a +-1 confidence interval on a bootstrap value of 95, 2000 replicates are necessary”.

You write “Other branch support methods exist” (L359) and then refer to the Shimodaira-Hasegawa test [124]. The bootstrap method is not a “branch support method”, so it is not correct to say “Other branch …”. Moreover, the Shimodaira-Hasegawa test is not a branch support test either. It is a test designed to compare the ML tree to other trees.

Response: We explain more in details the SH test and rewrote that part of the section: 

L432-437: “With ML-based phylogenetic inference, the Shimodaira-Hasegawa test, and its improved version AU [147], are designed to evaluate alternative phylogenetic hypotheses, and test if a tree is better supported than another one. It can be used with PAUP, PhyML, FastTree and IQ-TREE. The approximate likelihood ratio test [148] is another test of phylogenetic relationship, implemented in PhyML along with the bootstrapping method. Bayesian inference method calculates posterior probabilities, which measure branch support instead of bootstrap values.”

8. I have not included any specific comments on the two case studies, mainly because I have already used far more time on this review than I had available. That said, the two sections do need some revision.

9. None of the figures are as clear as is necessary for publication. In fact, I can’t read them.

The Bibliography is clearly set using a French application and do not appear to comply with PLoS One’s style. Please ensure that the Bibliography complies with the journal’s formatting style.

We modified the bibliography according to PLoS One`s style.

SPECIFIC COMMENTS

L25: “Here a” > “Here we present a”

Response: We changed “Here a” into “Here we present a”.

L26: “nucleic acid” > “DNA”

Response: We changed “nucleic acid” into “DNA”.

L27: Delete “is presented”

Response: We deleted “in presented”.

L33: “proto cells to modern organisms” > “primitive cells to modern cells and multicellular organisms,”

Response: We changed “proto cells to modern organisms” into “primitive cells to modern cells and multicellular organisms,”.

L36: “sequence composition of” > “information content in”

Response: We changed “nucleic acid” into “DNA”.

L38-L40: Poor sentence structure

L41: “mutations that” > “mutations (including substitutions) that”

Response: We changed “mutations that” into “mutations (including substitutions) that”.

L43: “all” > “both”

Response: We changed “all” into “both”.

L45: “past decades” > “past four decades”

Response: We added “four”.

L48: “taxa” > “taxa, genes or genomic components (e.g., transposable elements)”

Response: We added “, genes or genomic components (e.g., transposable elements)” in the sentence.

L52: “the others” > “other species”

Response: We changed “the others” into “other species”.

L53: “infection diseases” > “infectious diseases”

Response: We changed “infection” into “infectious”.

L59: Delete “both”

Response: We deleted “both”.

L65: Delete “the”

Response: We deleted “the”.

L66: “and of” > “as well as”

Response: We changed “and of” into “as well as”.

L67: “processes, following” > “processes. In so doing, we follow”

Response: We changed “processes, following” into “processes. In so doing, we follow”.

L68: “used by many and maintained” > “maintained and used by many”

Response: We changed “used by many and maintained” into “maintained and used by many”.

L75: Delete “peer-reviewed”

Response: We did not delete “peer-reviewed” because this exact sentence has to be included in the manuscript for protocols, according to the editor`s instructions for authors.

L80: Delete “digitalized

Response: We deleted “digitalized”.

L82: “nucleic acid or amino acid sequence” > DNA or amino-acid sequences”

Response: We changed “nucleic acid or amino acid sequence” into “DNA or amino-acid sequences”.

L88: Perhaps state that the protocol is based on that in [1].

Response: We added a reference to the new protocol (ref [1]) in the figure`s caption.

L89: “Nucleic acid” > “DNA”

Response: We changed “nucleic acid” into “DNA”.

L95: ”browser focusing on chordates that contains” > ”browser that focuses on chordates and contains”

Response: We changed “browser focusing on chordates that contains” into “browser that focuses on chordates and contains”.

L97: “This” > “The”

Response: We changed “This” into “The”.

L99” Delete “Overall,”

Response: We deleted “Overall”.

L105: “rendered” > “provided”

Response: We changed “rendered” into “provided”.

L114: “provides several” > “provides access to several”

Response: We changed “provides several” into “provides access to several”.

L116: “under” > “in the”

Response: We changed “under” into “in the”.

L117: “nucleotide sequences” > “nucleotide”

Response: We changed “nucleotide sequences” into “nucleotide”.

L118: Delete “pasted to fasta file”

Response: We deleted “pasted to fasta file”.

L129: “format and the” > “format. The”

Response: We changed “format and the” into “format. The”.

L129: “interlinks” > “links”

Response: We changed “interlinks” into “links”.

L130: “allows to batch download the” > “facilitates batch downloads of”

Response: We changed “allows to batch download the” into “facilitates batch downloads of”.

L132: “databases” > “databases (such as …)”

Response: We changed “nucleic acid” into “DNA”.

L136: “The KEGG focuses” > “The KEGG database focuses”

Response: We changed “nucleic acid” into “DNA”.

L148-L156: You alternate between “3-dimensional” and “3D”. Be consistent

Response: We wrote “3-dimensional” in the whole manuscript to homogenize.

L149: Delete “the”

Response: We deleted “the”.

L153: Place “i.e., … family” between parentheses

Response: We added parentheses.

L159: You write “based on …, or both”. Perhaps separate the databases accordingly.

Response: We agree the sentence was unclear. All three databases use structural similarity, functionality, and evolutionary relationship. We changed the sentence to make it more understandable: 

L162-163: “Proteins are classified into different categories based on structural similarity, functionality and evolutionary relationship (Table 2).”

L163: You write “based on the presence or absence of … strands”. I think you need to elaborate a bit further on this. There is SCOPe classification system to consider but you also need to consider the work by Sun et al. (Current Protocols in Protein Science (2004) 17.1.1-17.1.189)

Response: We added precisions on the structural classification.

L166-169: “Most proteins belong to one of the five main structural classes (α, β, α/β, α+β, multi-domain), defined respectively by the presence in the structure of α-helices, β-strands, both α-helices and β-strands, segregated α-helices and β-strands, or none of these characteristics.”

We added mention to two other classification systems:

L172-179: “Similarly, the Class, Architecture, Topology, Homology database (CATH) proposes a five-level classification of protein domains. The first three levels, namely class, architecture and topology, are based on structural homology. The last two, homologous superfamily and family, are based on sequence, structure and functional data, and sequence identity, respectively [29]. The Families of Structurally Similar Proteins (FSSP) provides a classification of proteins of the PDB based on a structure comparison algorithm, that calculate a structural similarity score between protein chains. These similarity scores are used to create a classification of protein structures [30,31].”

L164: “class” > “classification”

Response: We kept class instead of classification because we are talking about the protein “class” in the SCOPe system, not the protein classification. 

L177: “[16], that combines” > “[16]. It combines”

Response: We changed “[16], that combines” into “[16]. It combines”.

L186: Delete “whole”

Response: We deleted “whole”.

L188: “hemoglobine” > “haemoglobin”

L189: “hemoglobine” > “haemoglobin”

L190: “myoglobine” > “myoglobin”

Response: We changed “hemoglobin” haemoglobin” and “myoglobine” into “myoglobin”.

L197: Include a citation to the E-value. You might know it, but you are targeting beginners who can be assumed to be ignorant.

Response: We added a citation of Kerfeld and Scott, 2011, for the E-value.

L206: “retrieving from” > “retrieving sequences from”

Response: We changed “retrieving from” into “retrieving sequences from”.

L213: “related to each other” > “related to each other in an evolutionary sense” [they can be related to each other on other senses]

Response: We changed “related to each other” into “related to each other in an evolutionary sense”.

L220: “row one” > row of a matrix, one”

Response: We changed “row one” into “row of a matrix, one”.

L230-L231: Poor sentence structure and you probably mean “user interface”, rather than “web interfaces”

Response: We changed “web interface” into “user interface” and rewrote the sentence.

L233: “calculation times” > “execution times”

Response: We changed “calculation times” into “execution times”.

L236-L237: You need a reference to support this statement, and you then need to cite Golubchik et al. (Mol. Biol. Evol. 24, 2433–2442 [2007]), who reported the opposite for PRANK.

Response: We remastered the sentence on the specificity of Prank: L252: “PRANK is meant for closely related sequences [57].”

L238: “that” > “which”

Response: We changed “that” into “which”.

L242: “web interface” > “user interface”

Response: We changed “web interface” into “user interface”.

L253: “quantified and phylogenetic informative” > “quantified using AliStat [53] and phylogenetically informative”

Response: We changed “quantified and phylogenetic informative” into “quantified using AliStat [53] and phylogenetically informative”.

L258: “stating for example that” > “stating, for example, that”

Response: We changed “stating for example that” into “stating, for example, that”.

L259: “constant” > “constant over time (i.e., time-homogeneity)”

Response: We changed “constant” into “constant over time (i.e., time-homogeneity)”.

L260: “reversible” > “reversible and, therefore, also stationary (for details on these assumptions, see Jayaswal et al. (Syst. Biol. 63, 726-742 [2014]) and papers cited therein)”

Response: We changed “reversible” into “reversible and, therefore, also stationary (for details on these assumptions, see Jayaswal et al. (Syst. Biol. 63, 726-742 [2014]) and papers cited therein)”.

L261: You write “can be biased [58]”. It would be more appropriate to cite Syst. Biol. (53, 623-637; 53, 638-643) because these papers used data generated by simulation (hence, the truth is known).

Response: We added a citation of Syst. Biol. (53, 623-637; 53, 638-643).

L262: “recent phylogenetic protocols recommend … possible [1]” > “a recent phylogenetic protocol [1] recommends … possible”

Response: We changed “recent phylogenetic protocols recommend … possible [1]” into “a recent phylogenetic protocol [1] recommends … possible”.

L264: “developped" > “developed”

Response: We changed “developped” into “developed”.

L264: You write “tests for all these assumptions”. This is not true. Some have been included in IQ-TREE and IQ-TREE2. I agree that the matched-pairs tests of homogeneity (Bioinformatics 22, 1225-1231) are of relevance, but only for some Markovian conditions. One of these is implemented in Homo 2.1 (https://github.com/lsjermiin/Homo.v2.1).

Response: We changed “tests for all these assumptions” into “Statistical methods allowing to test stationarity, homogeneity under certain markovian conditions, and treelikeness”.

L266: “Selection of the molecular evolution model” > “Selection of the optimal model of sequence evolution”

Response: We changed “Selection of the molecular evolution model” into “Selection of the optimal model of sequence evolution”

L267: Several papers and book chapters have described model selection in general terms. I think it would be wise to cite some of these (e.g., Front. Genet. 6, 319 [2015; Meth. Mol. Biol. 1525, 379-420 [2017]; DOI 10.1007/978-1-4939-6622-6_15).

Response: We added references to those papers.

L268: “Nucleotide or amino acid substitution exist” > “Several models of nucleotide or amino-acid substitution exist [REF]”. The statement requires some citations.

Response: We changed “Nucleotide or amino acid substitution exist” into “Several models of nucleotide or amino-acid substitution exist” and added reference to the phylogenetic handbook (Yang, Oxford University Press, 2014).

L270: you cite Yang 2006; you should cite is more recent book from 2014.

Response: We added a reference to this book.

L273: You write “Each model … IQ-TREE [75]”. The set of models described here differ from the substitution models described in L271-L274. They are homotachous models of rate-heterogeneity across sites, so name them appropriately. You forgot to include the PDF/FreeRate model proposed by Yang (Genetics 139, 993-1005 [1995]) and now included in ModelFinder, IQ-TREE and PhyML.

Response: We rewrote the section to better separate the substitution models and the rate-heterogeneity models. We explain more in detail what rate-heterogeneity refers to. We also added reference to the FreeRate model and the relevant literature.

L331-338: “These substitution models can be associated with models of substitution rate-heterogeneity between sites. Mutation rates and selective pressure may vary among sites, due to different roles in the structure and function of the gene or protein. The most common rate heterogeneity model are the Gamma distribution (G) and the proportion of invariant nucleotide or amino acid sites (I). Every substitution model can be associated with G, I or both. The FreeRate model, a more complex model of rate heterogeneity [94], is included in ModelFinder, PhyML and IQ-TREE. More recently, the GHOST model for alignments with variation in mutation rate was introduced and implemented in IQ-TREE [95].”

L279: “every substitution model” > “model of sequence evolution (i.e., combination of substitution model and rate-heterogeneity across sites model)”

Response: We changed “every substitution model” into “model of sequence evolution (i.e., combination of substitution model and rate-heterogeneity across sites model)”.

L281: “optimizing” > “minimising”

Response: We changed “optimizing” into “minimising”.

L284: Move “PartitionFinder 2 …[81]” to L288. The logic here is that ModelFinder is for a single partition and it supersedes ModelTest, jModelTest and ProtTest. PartitionFinder 2 is for multiple partitions.

Response: We moved the mention of PartitionFinder2 to the end of the section.

L293: Replace the sentence with “Phylogenetic trees may consider the topology and the branch lengths (phylograms) or just the topology (cladograms)”

Response: We replaced the sentence with “Phylogenetic trees may consider the topology and the branch lengths (phylograms) or just the topology (cladograms)”.

L294: “tree building methods” > “tree-building methods” [compound adjective]

Response: We changed “tree building methods” into “tree-building methods”.

L295: “distance” > “distances”

Response: We changed “distance” into “distances”.

L295-L206: “number of differences” > “numbers and types of differences”

Response: We changed “number of differences” into “numbers and types of differences”.

L296: “to reconstruct” > “and they use this matrix to reconstruct”

Response: We changed “to reconstruct” into “and they use this matrix to reconstruct”.

L296-L297: The sentence is wrong. Distance methods are also character-based methods.

We left the distinction between distance methods and character-based methods, since this classification is common in reviews and books of the field (e.g. De Bruijn 2014, Muntjal el al 2019…).

L299: “classic” > “classical”

Response: We changed “classic” into “classical”.

L303: Include citation after “long branch attraction”

Response: We added a reference to the review by Bergsten, 2005.

L309: “mean” > “Mean”

Response: We changed “mean” into “Mean”.

L310: “[91].” > “[91] tree-inference methods.”

Response: We added “tree-inference methods.”

L312: “amino acid data” > “amino-acid data” [compound adjective]

Response: We changed “amino acid data” into “amino-acid data”.

L315: “but also” > “and”

Response: We changed “but also” into “and”.

L316: Delete “,”

Response: We deleted this comma.

L317: “in R language that” > “in the R language. It”

Response: We changed “but also” into “and”.

L320: “ML” > “maximum likelihood [ML]”. I think this is the first time that you typed ML, so it has to be defined, even though you and I both know what it refers to

Response: We left ML because the term has been defined earlier in the text (L361).

L322: “ML methods” > “Maximum-likelihood methods”. When starting a sentence, it is good practice to spell out the abbreviation (or number)

Response: We changed “ML methods” into “Maximum-likelihood methods”.

L322: “probability” > “likelihood”

Response: We changed “probability” into “Maximum-likelihood methods”.

L323: “ML aims” > “Maximum likelihood aims”

Response: We changed “ML” into “Maximum likelihood”.

L323: “combinations of model” > “combinations of trees and model”

Response: We changed “combinations of model” into “combinations of trees and model”.

L327: “amino acid data” > “amino-acid data”

Response: We changed “amino acid data” into “amino-acid data”.

L328: “is accurate” > “is reported as being accurate”

Response: We changed “is accurate” into “is reported as being accurate”.

L329: “all common models of molecular evolution” > “many common models of substitution” or “many common models of sequence evolution” [choose the most accurate statement”

Response: We changed “all common models of molecular evolution” into “many common models of sequence evolution”.

L331: “rate heterogeneity … invariant sites” > “rate-heterogeneity across sites (…)”. Here, you replace “…” with the model of that you refer to

Response: We changed the sentence and cite the method mentioned and relevant publication:

L396: “They use CAT, a specific model of rate heterogeneity, faster than G [129], in addition to Gamma law and proportion of invariant sites.”

L333: “that” > “which”

Response: We changed “that” into “which”.

L333: “method” > “method [REF]”. Here you need to replace “REF” with the relevant citation to the method implemented in IQ-TREE and IQ-TREE 2

Response: We added a reference to the paper by Hoang et al, 2017, describing UFBoot2, and mention the tool in the text.

L336: “recent method” > “recently-developed method”

Response: We changed “recent method” into “recently-developed method”.

L336: “inference, that” > “inference (BI). It”

Response: We changed “inference, that” into “inference (BI). It”.

L337: “calculates the probability” > “calculates the posterior probability”

Response: We changed “calculates the probability” into “calculates the posterior probability”.

L337: “the molecular evolution model given” > “the tree and model of sequence evolution, given”

Response: We changed “the molecular evolution model given” into “the tree and model of sequence evolution, given”.

L338: “ [109] that” > “ [109]. It” 

Response: We changed “[109] that” into “[109]. It”.

L362: “inbcludes” > “includes”

Response: We corrected the typo.

L365: “taxa” > “sequences”

Response: We changed “taxa” into “sequences”.

L366: “addressed question” > “question asked”

Response: We changed “addressed question” into “question asked”.

L370: “taxa” > “sequences”

Response: We changed “taxa” into “sequences”.

L370: “studied ingroup” > “ingroup of interest”

Response: We changed “studied ingroup” into “ingroup of interest”.

L373: “identified, … viruses,” > “included”

Response: We changed “identified, … viruses,” into “included”.

L375: “of the longest branches” > “between the most dissimilar sequences in the tree”

Response: We changed “of the longest branches” into “between the most dissimilar sequences in the tree”.

L378: “exported using … a graphical” > “visualised using a graphical”

Response: We changed “exported using … a graphical” into “visualised using a graphical”.

L394: You should include Geneious (https://www.geneious.com/) in the table and mention it in the text.

Response: We added Geneious in the table and in the text.

L397: “allows” > “allow users”

Response: We changed “allows” into “allow users”.

L404: Here you should only write MP, ML and BI. Note that you need to define MP somewhere earlier on in the text

Response: We wrote MP, ML and BI. MP is defined earlier in the text.

L405: “molecular” > “sequence”

Response: We changed “molecular” into “sequence”.

L416: In this section, you use the term “mutations”; typically the term “substitutions” is used

Response: We changed “mutations” into “substitutions” in the whole section.

L417: “The strength” > “The type and strength”

Response: We changed “The strength” into “The type and strength”.

L417: “can be calculated. This” > “may be of interest. It”

Response: We changed “can be calculated. This” into “may be of interest. It”.

L418: “ration” > “ratio”

Response: We corrected the typo.

L421: “The ratio … 1, that” > “If dN/dS > 1, then the”

Response: We changed “The ratio … 1, that” into “If dN/dS > 1, then the”.

L426: “Phylogenetic calibration” > “Molecular dating of events”

Response: We changed “Phylogenetic calibration” into “Molecular dating of events”.

L427: “event using events” > “events, using other events”

Response: We changed “event using events” into “events, using other events”.

L440-L444: This can be stated more succinctly

Response: We rewrote the sentences in a more understandable way to introduce coevolution.

L450: Why do you refer to Table 8 here?

Response: We moved the reference to Table 8 to the appropriate sentence.

L457: “species” > “traits”

Response: We changed “species” into “traits”.

L462-L466: This needs to be stated better

We rewrote the sentence:

L554-557: “Phylogenetic trees, in complement with genomics tools and databases, can be used to study genome evolution; and identify evolutionary events such as be mutations, insertions, deletions, gene or genome duplications, genome re-organization, chromosomal rearrangements, polyploidization events or genetic exchanges.”

L477: “Population genetics often studies” > Population geneticists often study”

Response: We changed “genetics” into “geneticists”.

L483: “of” > “at”

We changed “of” into “at”.

L484: “studies, for a full review see” > “studies; for a full review, see”

We changed “studies, for a full review see” into “studies; for a full review, see”.

L493: “in Å” > “in ångström”

We changed “in Å” into “in ångström”.

L494: You write “can be calculated”. Which program?

We added “using PyMol”.

L495: “3-dimentional” > “3D”

To be more consistent, we wrote “3-dimentional” in the whole manuscript.

L496: “amino acid” > “amino-acid”

We changed “amino acid” into “amino-acid”.

L498: Here I think you need to include Alpha fold (Nature 596, 583–589 [2021])

We included Alpha fold in the text and the table.

---

## [Decision Letter · Decision Letter 2]

12 Dec 2022

Roadmap to the study of gene and protein phylogeny and evolution – a practical guide

PONE-D-22-06715R2

Dear Dr. Hammarlund,

We’re pleased to inform you that your manuscript has been judged scientifically suitable for publication and will be formally accepted for publication once it meets all outstanding technical requirements.

Kind regards,

Arndt von Haeseler

Academic Editor

PLOS ONE

Additional Editor Comments (optional):

You have addressed all major comments. I trust you to make the minor correction suggested by the reviewer.

Otherwise the manuscript is now ready for publication.

Reviewers' comments:

Reviewer's Responses to Questions

**Comments to the Author**

1. Does the manuscript report a protocol which is of utility to the research community and adds value to the published literature?

Reviewer #2: Yes

2. Has the protocol been described in sufficient detail?

To answer this question, please click the link to protocols.io in the Materials and Methods section of the manuscript (if a link has been provided) or consult the step-by-step protocol in the Supporting Information files.

The step-by-step protocol should contain sufficient detail for another researcher to be able to reproduce all experiments and analyses.

Reviewer #2: Yes

3. Does the protocol describe a validated method?

Reviewer #2: No

4. If the manuscript contains new data, have the authors made this data fully available?

Reviewer #2: Yes

**5. Is the article presented in an intelligible fashion and written in standard English?**

Reviewer #2: Yes

6. Review Comments to the Author

Reviewer #2: GENERAL COMMENTS

The manuscript has improved substantially and is pretty much ready for publication.

Below is a list of minor issues that I suggest you attend to before returning the final version to the journal.

Lastly, well done on assembling so much knowledge. I might use your paper as reading material for 3rd-year biology students doing bioinformatics. Take that as a compliment!

SPECIFIC COMMENTS

L21: “into” > “of”

L61: “coupled to” > “coupled with” [I think this is the correct expression]

L69: “used by many” > “popular”

L86: Figure 1 is almost ready. However, the feedback loops are not accurate. You need to go back to the corresponding figure in Jermiin et al. (2020) and replicate those feedback loops.

L104: “of” > “on”

L118: “plant specific” > “plant-specific” [compound adjective, so it is hyphenated]

L119: What is the “efP” browser? It is listed without citation and URL. Should it be spelled out?

L123: “batch download” > “batch-download” [compound verb]

L129: Is “General information on proteins” a subheading under “Protein databases”? If so, then use a smaller font.

L131: “open reading frame translations” > “open-reading-frame translations” [compound adjective]

L138: You use the abbreviation “PDB” before you define it [line 153]. Please fix it.

L153: PDB should be defined earlier in the document and its citation should be moved to L138.

L154: “3-dimensional” > “three-dimensional (3D)”

L156: “three-dimensional (3D)” > “3D”

L158: “3-dimensional” > “3D”

L164: “The structural” > “The 3D structural”

L174: “database (CATH)” > “(CATH) database”

L178: “of” > “in”

L196: The definition of ‘homologues’ should be moved to where the one of the following three terms is first used (‘homology’, ‘homologous’, and ‘homologues’0) [perhaps L100; in fact, it may be sensible to define ‘homologues, orthologues, paralogues and zenologues in the Introduction, because these terms are used widely].

L217: “exhaustive research of homologues” > “exhaustive search for homologues” [I think this is what you mean]

L226: “amino acid” >“amino-acid”

L228: “tackle the tools these different steps” > “describe the tools used in these different steps”

L245: “] display user interfaces” > “] display inferred MSAs using/in user interfaces”

L247: “They” > “Their use”

L249: “for” > “to analyse”

L251: “closely related” > “closely-related”

L251: “Bayesian” > “Bayesian”

L253: “machine learning” > “machine-learning”

L271: You need more than nucleotide to make a sequence, so revise the sentence.

L286: You should include AliStat in this list.

L291: “Phylogenetic models” > “Phylogenetic methods”

L291: “simplifying assumptions stating” > “simple assumptions about the evolutionary processes, stating”

L295: “can be” > “may become”

L296: “sites selected” > “sites have been selected”

L297: “allowing to” > “allowing users to”

L298: “stationarity, homogeneity under certain Markovian conditions” > “stationarity and homogeneity of the evolutionary processes (along diverging lineages)”

L300: “)[“ > “) [“

L307: “, and most” > “. Most”

L311: “sequence evolution model” > “model of sequence evolution”

L313: “an inappropriate approximation” > “inappropriate approximations”

L325: Delete “substitution”

L326: “between” > “across”

L327: “rate heterogeneity models” > “rate heterogeneity across sites models”

L334: “combination of substitution model and rate-heterogeneity” > “a combination of a substitution model and a rate-heterogeneity”

L340: “for alignments … acids” > “, for alignments … acids, ”

L342: Delete “test”

L372: “analyses including” > “analyses, including”

L386: Do you mean “tree manipulators”?

L386: “of” > “to”

L389: You use the term “Gamma law”. I don’t like it, and suggest you call it something else (e.g., “to assuming Gamma-distributed rate-heterogeneity across sites”).

L394: “recently developed” > “recently-developed”

L398: “bayesian” > “Bayesian”

L399: “model well” > “model. It is well”

L412: “all informative characters” > “all resampled datasets”

L413: “few characters” > a few of these datasets”

L426: “version” > “version,”

L432: In this section, you should consult and cite Syst. Biol. 52, 229-238 and Mol. Biol. Evol. 24, 2400-2411. These papers are highly relevant.

L450: “in” > “is”

L456: “good” > “recommended”

L459: After the PP test, you should cite Syst. Biol. 63, 309-321.

L462: Delete “a”

L480: “3-dimensional” > “3D”

L494: IQ-TREE and IQ-TREE 2 also allow users to infer ancestral sequences, so it should also be listed here.

L512: In this section, you need to consult and cite Trends in Genetics 20, 80-86. It is extremely funny and highly relevant.

L548: You refer to “steps 1 and 2” in Figure 1, but they are not numbered, and the figure is very blurry (I cannot read the letters). You should fix this?

L575: “3-dimensional” > “3D”

L578: Great to see “ångström” spelled correctly.

L580: “amino acid” > “amino-acid

L597: “step by step” > “step-by-step”

L610: Move “instead” to after “but” [you used a Swedish sentence structure]

L613: “1765 … in Pfam” > Pfam contains 1765 P53 sequences from 382 species” [you never start a sentence with a number, and if you do have to, then that number must be spelled out]

L617: Sentence starts with a number [see previous comment]

L628: Sentence starts with a number [see previous comment]

L619: Sentence starts with a number [see previous comment]

L623: I cannot read Figure 2A. The colours make it impossible for me to see what you are trying to show. I am not colour-blind. Something about the figure needs to change.

L636: “3” > “Three”

L645: Is there a reason type-setting this paragraph in Italics? It looks as if it is an invitation to the reader. Should this not be made clearer, for example on L588?

L656: “has been used” > “was chosen”

L662: The text in this paragraph assumes a multiple sequence alignment involving all sequences in question. What you might have discussed percent identities based on pairwise alignments. I think you need to move this paragraph to after subsection “3. Multiple sequence alignment …”.

L668: Is there a reason type-setting this paragraph in Italics? It looks as if it is an invitation to the reader. Should this not be made clearer, for example on L588?

L679: “select” > “identify” [you can only select something after having identified it]

L691: Don’t write “Gamma law”. Find a more accurate term.

L694: “user interface or online” > “a user interface or an online”

L723: “[101] using” > “[101], using”

L723: “numbers indicate” > “numbers on the internal edges/branches indicate”

L762: Has anyone publish an evolutionary history of P53 before this submission. If so, then you could add a sentence or two about this and about how your result compares to the previous one.

L780: “21” > “Twenty-one”

L789: “has been” > “was”

L792: “tested with” > “consistency of the phylogenetic estimate was assessed using”

7. PLOS authors have the option to publish the peer review history of their article (what does this mean?). If published, this will include your full peer review and any attached files.

Reviewer #2: No

---

## [Editor Report · Acceptance letter]

22 Dec 2022

PONE-D-22-06715R2 

Roadmap to the study of gene and protein phylogeny and evolution – a practical guide 

Dear Dr. Hammarlund:

I'm pleased to inform you that your manuscript has been deemed suitable for publication in PLOS ONE. Congratulations! Your manuscript is now with our production department. 

Kind regards, 

on behalf of

Prof. Arndt von Haeseler 

Academic Editor

PLOS ONE